# Temporal dynamics of climate change exposure and opportunities for global marine biodiversity

Andreas Schwarz Meyer [1] ✉, Alex L. Pigot [2], Cory Merow [3], Kristin Kaschner [4], Cristina Garilao[5], Kathleen Kesner-Reyes [6] & Christopher H. Trisos[1,7] ✉

Climate change is exposing marine species to unsuitable temperatures while also creating new thermally suitable habitats of varying persistence. However, understanding how these different dynamics will unfold over time remains limited. We use yearly sea surface temperature projections to estimate temporal dynamics of thermal exposure (when temperature exceeds realised species' thermal limits) and opportunity (when temperature at a previously unsuitable site becomes suitable) for 21,696 marine species globally until 2100. Thermal opportunities are projected to arise earlier and accumulate gradually, especially in temperate and polar regions. Thermal exposure increases later and occurs more abruptly, mainly in the tropics. Assemblages tend to show either high exposure or high opportunity, but seldom both. Strong emissions reductions reduce exposure around 100-fold whereas reductions in opportunities are halved. Globally, opportunities are projected to emerge faster than exposure until mid-century when exposure increases more rapidly under a high emissions scenario. Moreover, across emissions and dispersal scenarios, 76%-97% of opportunities are projected to persist until 2100. These results indicate thermal opportunities could be a major source of marine biodiversity change, especially in the near- and mid-term. Our work provides a framework for predicting where and when thermal changes will occur to guide monitoring efforts.

Although climate change is affecting biodiversity on land and in the ocean, marine organisms have been found to be particularly sensitive to warming[1–3]. For instance, marine ectotherms tend to fully occupy their potential latitudinal range, indicating that their range boundaries closely track changes in temperature[1,2]. Compared to terrestrial ectotherms, marine ectotherms are also more susceptible to physiological stress from climate change due to their narrower thermal safety margins (that is, the smaller difference between body temperatures and the upper critical thermal limit of a species)[3]. Such sensitivity, coupled with fast climate change[4] and lower constraints on dispersal, has driven range shifts towards newly suitable habitats[5]. As a result, rates of local extirpation, colonisation, species richness change, and temporal

[1]African Climate and Development Initiative, University of Cape Town, Cape Town, South Africa. [2]Centre for Biodiversity and Environment Research, Department of Genetics, Evolution and Environment, University College London, London, UK. [3]Department of Ecology and Evolutionary Biology and Eversource Energy Center, University of Connecticut, Storrs, CT, USA. [4]Department of Biometry and Environmental Systems Analysis, Albert-Ludwigs University, Freiburg im Breisgau, Germany. [5]GEOMAR Helmholtz-Centre for Ocean Research, Kiel, Germany. [6]Quantitative Aquatics, Los Baños, Philippines. [7]African Synthesis Centre for Climate Change Environment and Development (ASCEND), University of Cape Town, Cape Town, South Africa. ✉e-mail: andreas.schwarzmeyer@uct.ac.za; christopher.trisos@uct.ac.za

turnover in community composition are higher for heating in the oceans than on land[3,5–7]. However, these changes are not randomly distributed in space and time. While at the equatorward edge of species ranges climate change is causing abundance declines and local extirpations, abundance increases and the influx of new species have been observed at higher latitudes[7–9]. Given the effect of temperature in determining both high- and low-latitude range limits of marine species[1,9,10], equatorward shifts can be strongly influenced by the loss of climatically suitable habitats and poleward shifts by thermal opportunities arising in previously unsuitable regions.

Numerous studies estimating future risks for marine biodiversity −by assessing how climate change will alter the thermal seascape and redistribute currently suitable habitats−project significant changes in the composition of communities, mainly due to an increasing rate of both species losses in the tropics and species invasion at higher latitudes[11–16]. Although these studies offer valuable insights into the magnitude of future marine biodiversity changes, they are often limited to one or two specific time points, typically towards the end of this century. Therefore, they lack the temporal perspective needed to answer fundamental questions such as when and how fast these changes will occur, leaving an important gap in our understanding of how changes in the thermal environment that constrain marine biodiversity will unfold over time.

Recent studies have started to examine temporal dynamics of species exposure to potentially unsafe temperatures using yearly or daily climate projections[17–20]. However, much less attention has been given to the temporal dynamics of the opposite process whereby climate change creates new thermally suitable habitats for species. Thus, we know little about when, where and how abruptly these 'thermal opportunities' will arise over the coming decades, and their similarities and differences to the spatio-temporal dynamics of thermal exposure. Understanding the temporal dynamics of thermal opportunities is important for several reasons. The impacts of climate change on a species are likely to depend in part on a balance between the rate and order in which thermally suitable habitats are lost and gained[21]. The emergence of new thermal opportunities may provide resilience to species by enabling them to expand their distributions to new locations, but only if those opportunities arise sufficiently early relative to thermal exposure, and only if they persist for long enough to enable successful colonisations and range expansions[9,21]. On the other hand, the emergence of thermal opportunities for one species can pose risks to others due to the potential for novel biotic interactions that disrupt local assemblages and drive local extinctions[22,23]. Better understanding the risks of species range contractions and novel biotic interactions is thus likely to depend in part on the relative timing of thermal exposure and opportunities.

Here, we model the future temporal dynamics of thermal exposure and opportunity for marine species assemblages globally. A total of 21,696 species were used with five phyla contributing 92% of the species: chordates (44%, mainly fish), molluscs (22%), arthropods (16%), cnidarians (6%) and echinoderms (4%) (Supplementary Fig. 1). Using yearly sea surface temperature projections from nine climate models and realised thermal niche limit estimates we quantified where and when thermally suitable habitats are created (thermal opportunities) and lost (thermal exposure) across low (SSP1-2.6), intermediate (SSP2-4.5) and high (SSP5-8.5) greenhouse gas emission scenarios up to 2100. Thermal exposure (hereafter 'exposure') occurs when a species in a site (that is, a 100 km grid cell in our analysis) is exposed to temperatures beyond its realised thermal niche for at least five consecutive years[17,19]. A thermal opportunity (hereafter 'opportunity') arises when the temperature at a previously unsuitable site near the existing range of the species becomes suitable for at least five consecutive years (that is, the temperature falls within the realised thermal niche limits of that species). Sea surface temperature is an important driver of marine species distributions, range shifts and community

turnover under climate change[24–29] and thus provides valuable insights into current and future climate change impacts, but does not fully capture changes in the deep ocean and potential for species to migrate to deeper waters. Our aim here is to model exposure and opportunity dynamics in the epipelagic layer of the ocean, with 88% of the species in our dataset occurring in the depth range 0–50 m and species occurring exclusively below 200 m excluded from our analysis. It is not our goal in this paper to predict the dynamics of biodiversity, which depends on multiple ecological (e.g. dispersal, predation and competition) and evolutionary (e.g. genetic adaptation) processes that remain poorly understood[9]. Instead, our aim is to model the dynamics of changes in the thermal environment available to marine species that will ultimately drive and constrain biodiversity responses.

To describe the temporal dynamics of both opportunity and exposure, as well as their interplay, we built on the approach of[17] to develop new metrics that apply to both thermal opportunity and exposure within a single framework. For each of 41,220 assemblages (that is, species in 100 km grid cells), we calculated: (i) magnitude of exposure, the proportion of species exposed locally; (ii) magnitude of the opportunity, the number of opportunities that arose locally (calculated as a proportion of the local species richness); (iii) timing of exposure, the median year in which exposure occurs in an assemblage; (iv) timing of the opportunity, the median year in which opportunities arise in an assemblage; (v) abruptness of exposure, the proportion of species exposed within the decade of maximum exposure; and (vi) abruptness of opportunity, the proportion of opportunities within the decade where the maximum number of opportunities arose. Additionally, we calculated how long opportunities are projected to persist. To avoid projecting opportunities in regions that are either environmentally unsuitable or unreachable, we constrained the grid cells where opportunities could arise by taking into account species' depth preferences and niche unfilling (when species fail to fully occupy specific locations within their thermal niche). To further constrain where opportunities could arise we used an average dispersal rate estimate of 10 km year[−1], based on dispersal data from empirical studies[5,6,30]. To test the influence of the dispersal rate on our projections, we repeated our analyses using a 50 km year[−1] rate, which is roughly the maximum dispersal rate for some marine organisms[5,6,31].

We show that thermal opportunities arise earlier and accumulate gradually, while thermal exposure increases later and more abruptly. Assemblages tend to experience either high exposure or high opportunity, but rarely both. Once they arise, most opportunities are projected to persist until 2100. Reducing greenhouse gas emissions has a greater effect on exposure. Our study highlights the potential of thermal opportunities to drive marine biodiversity changes, especially in the near- and mid-term, and offers a framework for predicting thermal seascape changes that can be used to develop early warning systems and guide monitoring efforts.

## Results and discussion
### Exposure and opportunity profiles
Exposure and opportunity profiles show distinct spatial and temporal patterns. While exposure is generally concentrated in the tropics, most opportunities arise at higher latitudes (Fig. 1a–c). Regions projected to experience the highest magnitude change−that is, opportunity or exposure for >10% of species richness in an assemblage−show either high exposure or high opportunity, but seldom both (Fig. 1a–c). For regions with high exposure, this suggests that any resulting local extirpations would generally lead to declines in species richness because limited opportunities could prevent colonisers from compensating for potential species losses where exposure is high. Conversely, many regions with lower exposure tend to have a higher concentration of opportunities. This could lead to a net increase in species richness in these regions in future, but also does not imply these regions are lower risk because the risk of ecological disruption

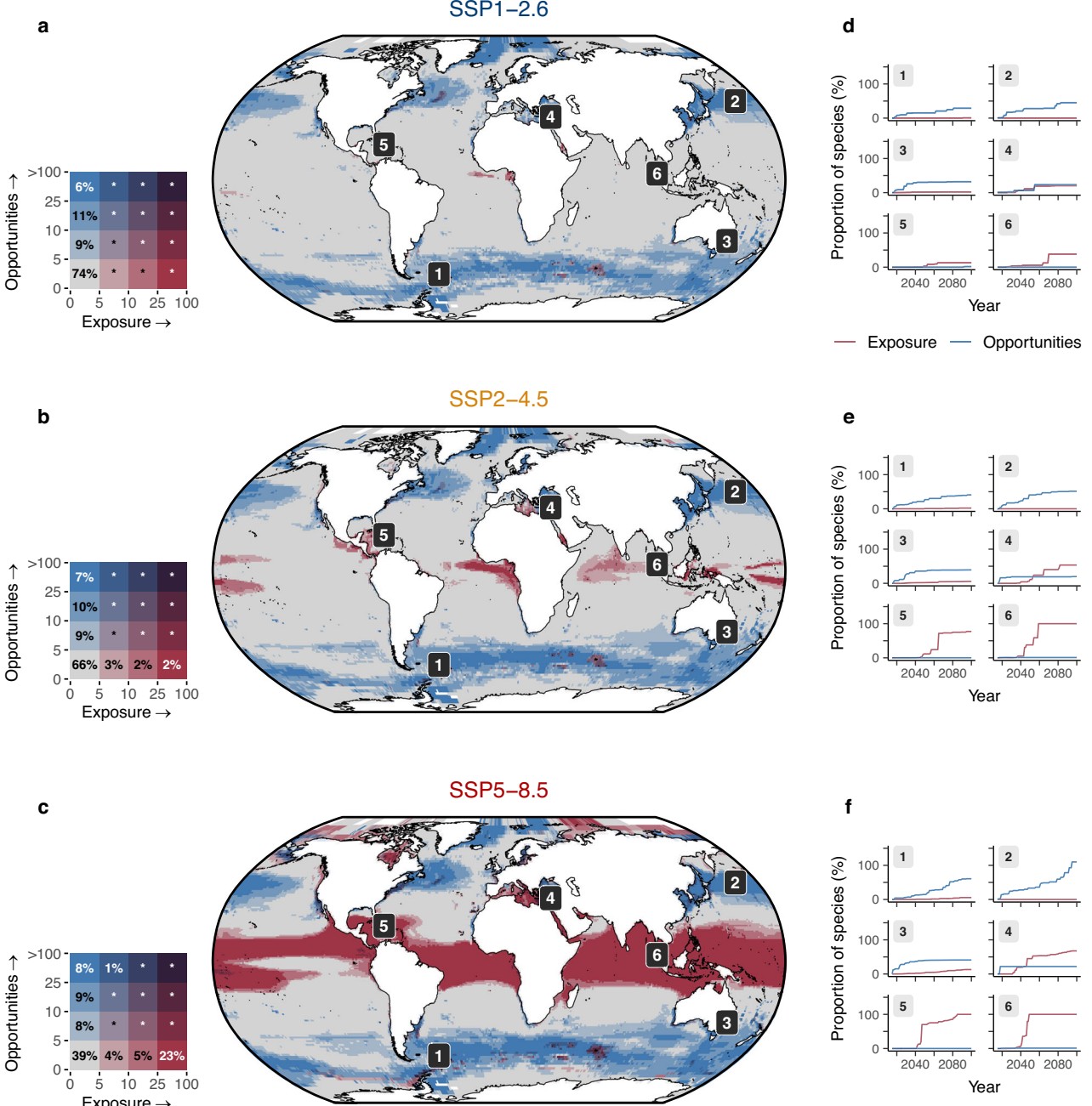

**Fig. 1 | The projected magnitude of thermal exposure and opportunity for marine biodiversity show distinct spatial patterns. a–c** Bivariate maps showing the proportion of species exposed (x-axis) and opportunities created (y-axis) as a percentage of local species richness across low (SSP1-2.6), intermediate (SSP2-4.5) and high (SSP5-8.5) emission scenarios. The percentages inside the key indicate the proportion of assemblages (that is, species in 100 km grid cells) within each bivariate bin. Asterisks indicate values below 1%. Lowering emissions from SSP5-8.5 (**c**) to SSP1-2.6 (**a**) has a greater impact on reducing exposure than on opportunities. **d–f** Examples of exposure and opportunity profiles for local assemblages for each emission scenario. Profiles correspond to the scenario represented on the map adjacent to the plots. Exposure was estimated only for native species. Opportunities are concentrated in temperate and polar regions, while exposure occurs mostly in the tropics. The profiles show how opportunities accumulate more gradually (sites 1–4) while exposure can be abrupt (sites 5 and 6). Most regions show either high opportunity or high exposure, although regions such as the Mediterranean (4) can show both. The figure shows the median value across nine climate models. Source Data for this figure can be found in ref. 72.

from new migrants may be elevated. Thermal exposure and opportunity together are projected to affect more than 10% of current species richness for 26% (SSP1-2.6), 34% (SSP2-4.5) and 61% (SSP5-8.5) of assemblages across the world's oceans (Fig. 1a–c).

Reducing future greenhouse gas emissions has a greater impact on reducing thermal exposure than opportunities in the upper layer of the ocean. When contrasting high- and low-emission scenarios, the magnitude of opportunities is halved, while the reduction in exposure is around 100-fold (Fig. 2a). The number of assemblages projected to have at least 10% of species exposed is projected to decrease from 28% under high emissions (SSP5-8.5) to <1% under low emissions (SSP1-2.6) (Fig. 1a, c). In contrast, for all emissions scenarios analysed, 17% of assemblages are projected to have opportunities corresponding to at least 10% of their current species richness (Fig. 1a–c). This finding

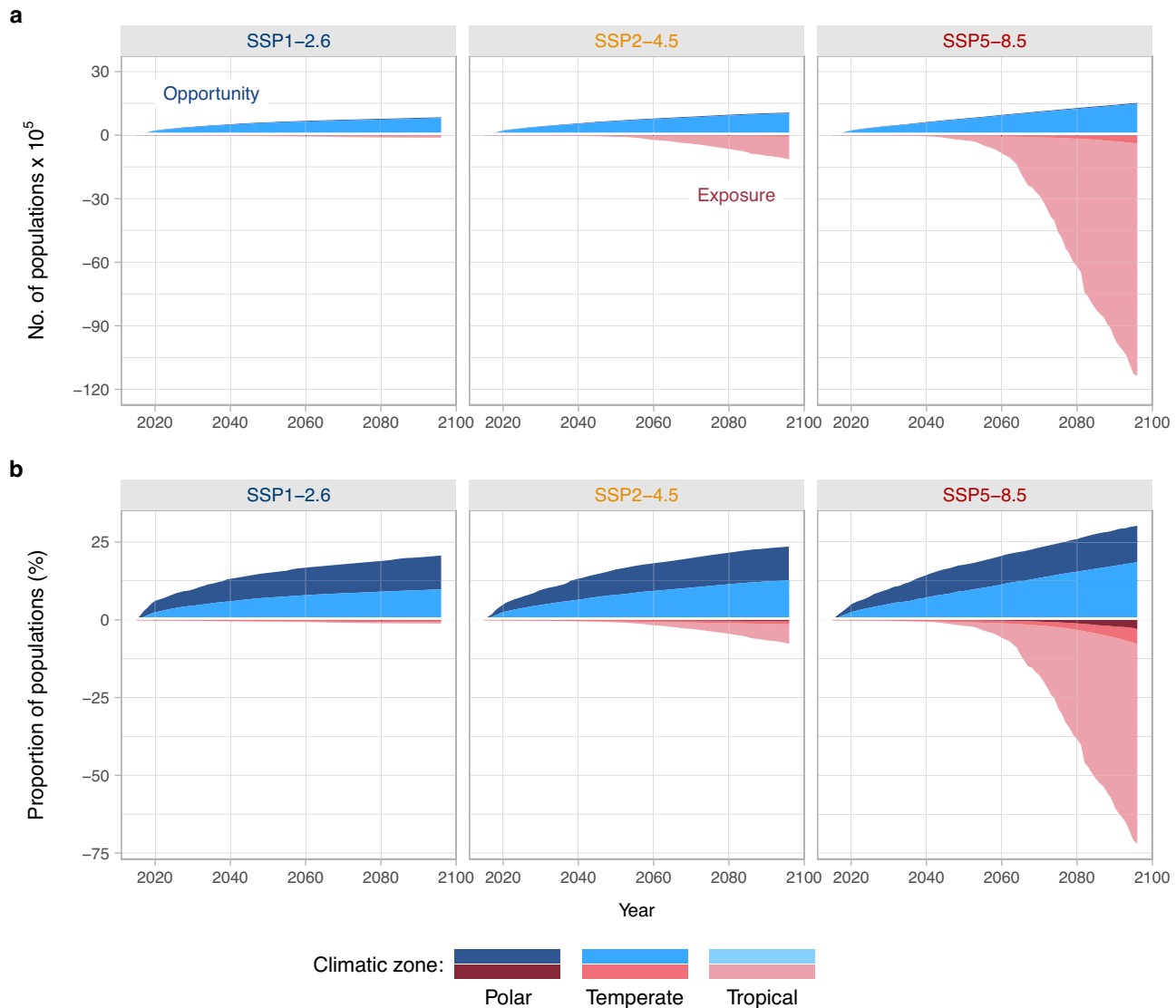

**Fig. 2 | Global profiles of thermal exposure and opportunities for marine biodiversity show earlier emergence of opportunities. a** The cumulative number of populations exposed (red) and opportunities (blue) over time across low (SSP1-2.6), intermediate (SSP2-4.5) and high (SSP5-8.5) emission scenarios. A population is defined as a species occurrence in an assemblage (i.e. grid cell). The different colour shades indicate the climatic zone where exposure and opportunity are projected (polar, temperate, or tropical zones). **b** Same data as in (**a**), but with exposure and opportunity shown as a proportion of the current number of populations within each climatic zone. Opportunities arise early in the century and follow a similar trajectory across scenarios until 2040. Exposure starts later and is substantially lower under SSP1-2.6 and SSP2-4.5 when compared to SSP5-8.5. The panels show the median value across nine climate models. Source Data for this figure can be found in ref. 72.

remains robust when dispersal is increased from 10 to 50 km year⁻¹ (Supplementary Fig. 2). While the magnitude of opportunities increases, the projected proportion of assemblages with opportunities corresponding to at least 10% of current species richness remains similar across emissions scenarios (42%, 47%, and 53% under SSP1-2.6, SSP2-4.5 and SSP5-8.5, respectively).

Spatio-temporal patterns of opportunity and exposure are largely consistent across phyla, with opportunities projected to arise mainly in temperate and polar regions and exposure to occur in the tropics (Supplementary Figs. 3, 4). Chordates account for the highest proportion of the projected exposures and opportunities in most grid cells, followed by molluscs and arthropods (Supplementary Figs. 3–5). In general, taxa that make up a higher proportion of the species richness in an assemblage contribute more to the magnitude of opportunities and exposure (Supplementary Fig. 5).

In absolute numbers, most thermal opportunities will emerge in the temperate zone (Fig. 2a). However, when the current diversity of

the region is taken into account, the poles are expected to undergo a greater proportional change in opportunities (Fig. 2b), especially under higher dispersal rates (Supplementary Fig. 6). A higher effect of opportunities at the poles aligns with previous research indicating higher levels of species turnover in these regions[11,12,16], mostly driven by net species gains.

Although earlier studies have shown that suitable habitats will disappear from the tropics and shift towards higher latitudes[11,12,16], our temporal approach provides a perspective on the timeline of these changes. We found that, across all greenhouse gas emission scenarios, opportunities begin to arise immediately at the start of the simulation (i.e. 2015) and follow a similar trajectory of gradual increase until around 2040 (Fig. 2). After this point, opportunities continue to emerge gradually throughout the century, but with higher greenhouse gas concentrations resulting in more opportunities. These findings indicate that the emergence of opportunities is a process already underway, consistent with the recent climate-driven range shifts observed in many

marine species[5,6,8,30]. The early emergence of thermal opportunities across all scenarios also suggests that for many species temperature might not be the main abiotic constraint for poleward range shifts in the near future. Consequently, the role of other abiotic factors such as oxygen and pH in constraining poleward range shifts may become more relevant in the next two decades[32,33]. Understanding how these factors will change in the future is likely to improve projections of near to mid-term range shifts in the epipelagic layer of the ocean.

In contrast to opportunities, exposure usually starts later and occurs more abruptly (Fig. 2). Until approximately 2060, global exposure levels exhibit a comparable pattern across all scenarios (Fig. 2). Beyond 2060, global exposure accelerates under SSP2-4.5 when compared to SSP1-2.6, reaching a similar number of opportunities by 2100. Under SSP5-8.5, exposure is projected to dramatically increase, exceeding the number of opportunities by more than seven times by the end of the century (Fig. 2). Previous studies that concentrated on a limited number of future time points have reported that under RCP8.5 species could lose suitable habitats at lower latitudes faster than the rate at which they will be gained at higher latitudes[16]. Our analysis, however, projects that the rate at which opportunities arise is consistently higher than the rate of exposure until around 2060, regardless of the emissions scenario analysed. This finer temporal resolution suggests for the near- and mid-term that opportunities are likely to be the primary factor driving changes in the thermal seascape in the coming decades outside the tropics.

The finding that opportunities are emerging earlier than exposure can be explained by the fact that the latitudinal range limits of marine ectotherms closely match their thermal limits[1,2]. Therefore, as soon as isotherms start to shift polewards, opportunities begin to arise. Such early opportunities may help to explain why range expansions have been reported as faster and more common than range contractions in the oceans[6,8,9]. Previous studies have provided various explanations for these observations, including stronger regional warming at the poleward edge of species ranges and long lag time to local extinction[34]. Our finding that thermal opportunities generated by climate change emerge earlier than the later and abrupt thermal exposure provides an additional, non-exclusive explanation for these ongoing temporal dynamics. In other words, patterns of range shifts across marine species may also reflect abiotic constraints imposed by the timing and magnitude of thermal exposure and opportunity, rather than only differences in the biological processes operating at leading and trailing range margins.

## Abruptness and timing of exposure and opportunity

Exposure and opportunities show distinct patterns of abruptness. While exposure becomes more abrupt with higher greenhouse gas emissions (Fig. 3a–c), the abruptness of opportunities shows a contrasting trend, with decreasing abruptness of opportunities for higher emissions scenarios (Fig. 3d–f). The higher abruptness of opportunities under SS1-2.6 reflects a greater proportion of opportunities arising before 2050 and limited opportunities arising later in the century as the rate of warming slows (Fig. 2). In contrast, the higher abruptness of exposure under SSP5-8.5 reflects the marked increase in exposure after 2060 with increasing warming (Fig. 2).

The tropics generally experience more abrupt changes in both exposure and opportunity (Fig. 3a–f), but there is substantial variability across assemblages globally and abrupt changes are also observed at higher latitudes, especially in the Northern hemisphere. Notably, the abruptness of opportunities in the tropics is likely influenced by the small number of opportunities projected to arise in these regions (see Fig. 2)−when only a few opportunities arise in an assemblage, less variation in timing is expected, leading to a higher abruptness.

The timing of exposure and opportunity follows a similar dynamic to each other, with more exposure and opportunity events projected to occur later in higher emissions scenarios as climate change

increases (Fig. 3g–l). However, many assemblages are projected to still experience an early emergence of opportunities under SSP5-8.5, especially in the tropical-temperate transition zone, the South Pacific Ocean, and the Arctic Ocean (Fig. 3l). The timing of exposure also shows considerable variability across assemblages under SSP1-2.6 (Fig. 3g) and SSP2-4.5 (Fig. 3h). Assemblages predicted to experience early exposure are generally found at both low and higher latitudes, while those predicted to experience later exposure are mostly concentrated in the tropics. This is due to abrupt increases in the number of species exposed in the tropics and subtropics as warming increases later in the century, pushing the median year of exposure later for tropical assemblages. Higher variability in the timing of abruptness or exposure across assemblages reflects lower abruptness because when exposure and opportunity events occur more gradually, median estimates of timing show greater variability. Patterns of timing and abruptness remain similar when a 50 km year$^{-1}$ dispersal rate is considered (Supplementary Fig. 7).

The timing and abruptness projections demonstrate how different climate change scenarios affect exposure and opportunity in different locations and at different times. While these results reinforce that reducing greenhouse gas emissions will have a stronger impact on reducing exposure, they also demonstrate that the effects on opportunities are more complex than just a reduction in magnitude. If humanity manages to rapidly reduce greenhouse gas emissions, fewer populations will experience thermal exposure, and the main changes in the epipelagic layer will be associated with opportunities arising in temperate and polar regions during the first half of this century. However, if emissions remain uncontrolled, exposure will be dramatically higher, and many marine assemblages will face a high risk of near-simultaneous exposure for more than 80% of the resident species, especially in the Indian and Pacific Oceans (Fig. 3c). With high emissions, opportunities will also continue to arise during the whole century, which will increase the risks to resident species from new biotic interactions, especially for temperate and polar assemblages.

## Persistent and transient opportunities

Most thermal opportunities arising this century are projected to persist beyond 2100 (Fig. 4a). Persistent opportunities represent 79, 91 and 97% of all opportunities projected to arise under SSP1-2.6, SSP2-4.5 and SSP5-8.5, respectively. Higher dispersal rates increase the number of opportunities by three to four times, yet the proportion of persistent opportunities remains very similar (Supplementary Fig. 8). Previous studies using climate velocity demonstrated that residence times of a given temperature climate (that is, the amount of time specific climate conditions persist across a given area under climate change) can be very short in temperate and polar biomes[35,36]. Based on these findings, one could expect that opportunities would quickly disappear as the climate continues to rapidly warm. However, our findings suggest that when sites become thermally suitable they are projected to remain so until at least the end of the century. This difference from previous results is likely to be explained by the fact that our model is based on when species thermal limits are exceeded, and these will typically be broader temperature climate ranges than when using only local climate variation as is the case in climate velocity estimates.

Under SSP1-2.6 and SSP2-4.5, most transient opportunities result from temperature drops below thermal limits (cold exposure). Conversely, the exceeding of thermal limits (warm exposure) caused most transient opportunities under SSP5-8.5 (Fig. 4b). Under higher dispersal, cold exposure was the leading cause of transient opportunities across all scenarios (Supplementary Fig. 8). The prominent role of cold exposure in creating transient opportunities can explain their higher frequency under lower emission (SSP1-2.6; Fig. 4a), where the noise to signal ratio of the long-term warming trend is weaker. These findings suggest that transient thermal opportunities in the oceans can arise from two distinct processes. While those projected to close due to

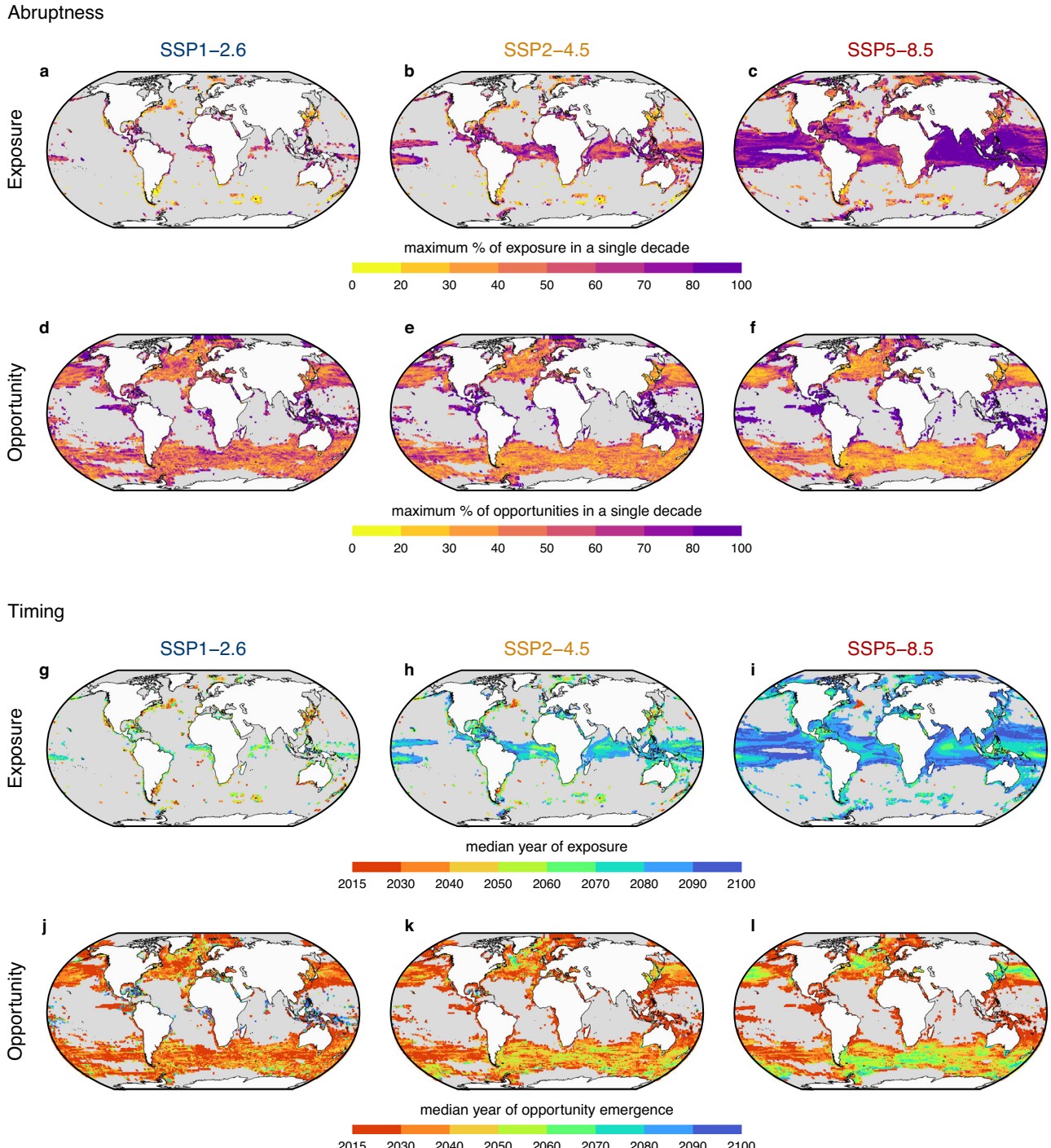

**Fig. 3 | Global variation in the abruptness and timing of exposure and opportunity.** Each column shows results from a different emission scenario. **a**–**c** Abruptness of exposure. **d**–**f** Abruptness of opportunity. **g**–**i** Timing of exposure. **j**–**l** Timing of opportunity. Exposure occurs more abruptly than opportunity under SSP5-8.5. Opportunities arise earlier and more abruptly under SSP1-2.6 when compared to SSP5-8.5 and SSP2-4.5. The maps show the metrics calculated from the median across nine climate models. Only assemblages with more than five species exposed or five opportunities are shown. Source Data for this figure can be found in ref. 72.

warm exposure reflect long-term trends in temperature change, those closed due to cold exposure are affected by short-term climate variability, which can produce brief periods of low temperatures even under a warming trend. As a result, transient opportunities tend to reopen after cold exposure, but not after warm exposure (Supplementary Fig. 9).

Whether transient opportunities will disappear due to warm or cold exposure might also have different implications for future biodiversity change. For example, warm-exposed transient opportunities are those that may serve as stepping stones for range shifts as isotherms move poleward[21]. These opportunities will generally arise early in the century (Fig. 5c) and be mostly concentrated in the tropics and in the North Atlantic Ocean (Fig. 4c). On average, the duration of warm-exposed transient opportunities is longer under SSP5-8.5 compared to lower warming scenarios (Fig. 5d). The longer duration of warm-exposed transient opportunities, combined with a greater

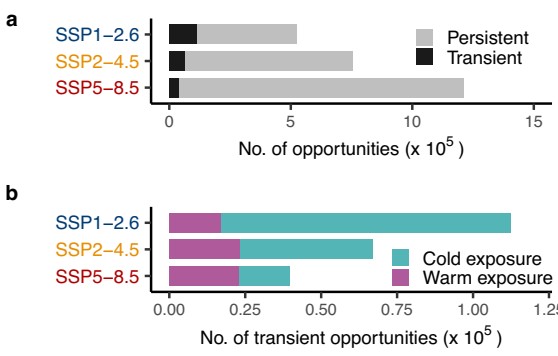

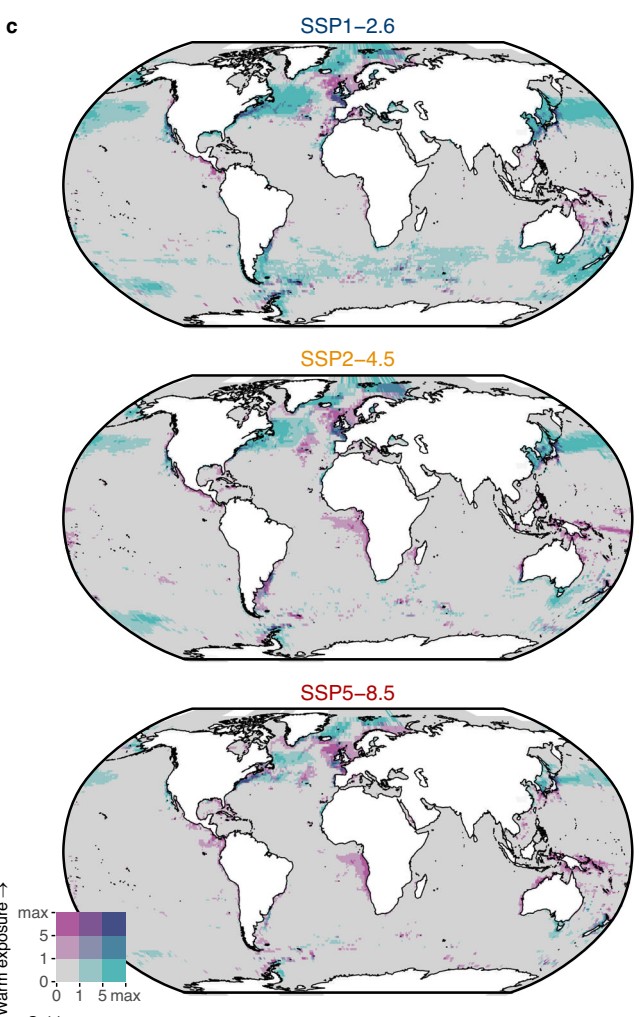

**Fig. 4 | Frequency and distribution of persistent and transient opportunities.**
**a** Frequency of persistent and transient thermal opportunities. **b** Frequency of
transient opportunities closed by exposure to temperatures above (warm expo-
sure) and below (cold exposure) the realised thermal niche limits of the species.
**c** Bivariate maps showing the geographical distribution of the median number of
warm- and cold-exposed opportunities for three emission scenarios. Transient
opportunities closed due to warm exposure are mostly concentrated in the tropics
and in the North Atlantic Ocean, while those closed due to cold exposure are
concentrated in temperate and polar regions, especially in the northern hemi-
sphere. The key indicates the number of transient opportunities. All figures
show the mean value across nine climate models. Source Data for this figure
can be found in ref. [72].

number of persistent opportunities, may further amplify their ecolo-
gical impacts especially if emissions remain unabated.

Transient opportunities closed due to cold temperatures could
result in a temporary increase in local biodiversity[37], followed by a
decrease in local abundance or a retraction of the species distribution
once temperatures fall below the realised niche limits of the species.
These opportunities are mostly found in temperate and polar regions
(Fig. 4c), and are projected to arise later and last longer under SSP1-2.6
and arise earlier and close sooner under SSP5-8.5 (Fig. 5a, b). Despite
their shorter duration compared to opportunities closed due to
warming (Fig. 5b, d), cold-exposed transient opportunities might
increase the risk of mortality of migrants[38], indicating their potential to
limit range expansion.

## Implications of exposure and opportunity

Marine species are rapidly shifting their geographical distributions in
response to changes in the thermal seascape[5,6,22,24,30]. As a result, many
ecological assemblages are experiencing species gains, losses, or a
combination of both. Our findings demonstrate that the emergence of
thermal opportunities on the ocean surface—a precursor of the colo-
nisation of new habitats—is a process already underway and is expec-
ted to consistently exceed thermal exposure in magnitude, except
under high emission scenarios. In contrast, while exposure is also
already occurring, it is projected to grow rapidly in magnitude globally
only after mid-century. These findings match the trend observed in
empirical studies that show that marine species are shifting the pole-
ward edge of their ranges faster than the equatorward edge[6,8].

The high number of opportunities projected to arise have
numerous implications for biodiversity. For example, if thermal
opportunities are more widespread than exposure as might be the case
under lower warming scenarios (Figs. 1, 2), then polewards range shifts
and range expansions are expected to be more common. Opportu-
nities could be important to prevent overall range contractions for
species that are unable to tolerate thermal changes across their native
range. The colonisation of a new location can thus increase resilience
and decrease global extinction risk for those species. The higher the
number of new opportunities projected to arise in the future, the
greater the potential for a larger number of species to benefit from
them. Humans may also benefit when new opportunities are projected
to increase the distribution range and abundance of species relevant to
local economies and food security[39–41]. However, for the organisms
living in recipient assemblages, opportunities for immigrating species
might mean increased risks. The influx of new species can introduce
new predators, pathogens and competitors, disrupting trophic inter-
actions and causing the decline of resident species[22]. The fact that
most opportunities will arise in areas with low exposure of resident
species suggests that low exposure does not imply low risk from cli-
mate change, as opportunities could cause significant impacts in
those regions.

The emergence of an opportunity does not necessarily translate
into biodiversity change, as species need to first reach and then suc-
cessfully colonise a site[42]. Therefore, the dispersal ability of the species
directly affects the likelihood of a thermal opportunity turning into a
successful colonisation. Our approach focused on projecting the
future dynamics of the emergence of thermal opportunities, including
their persistence or transience over this century, to better understand
how thermal opportunities arise within the region where a species
could potentially disperse to this century. We, therefore, took a more
permissive approach to identify such thermal opportunities within a
dispersal-constrained buffer zone, rather than a more restrictive
approach with annual dispersal increments, because this allows us to
identify the temporal dynamics of opportunities that may arise far
from the current range edge, but could be reached by rare long-
distance dispersal events or when the opportunities persist for several
decades. Our analyses demonstrated that higher dispersal predictably

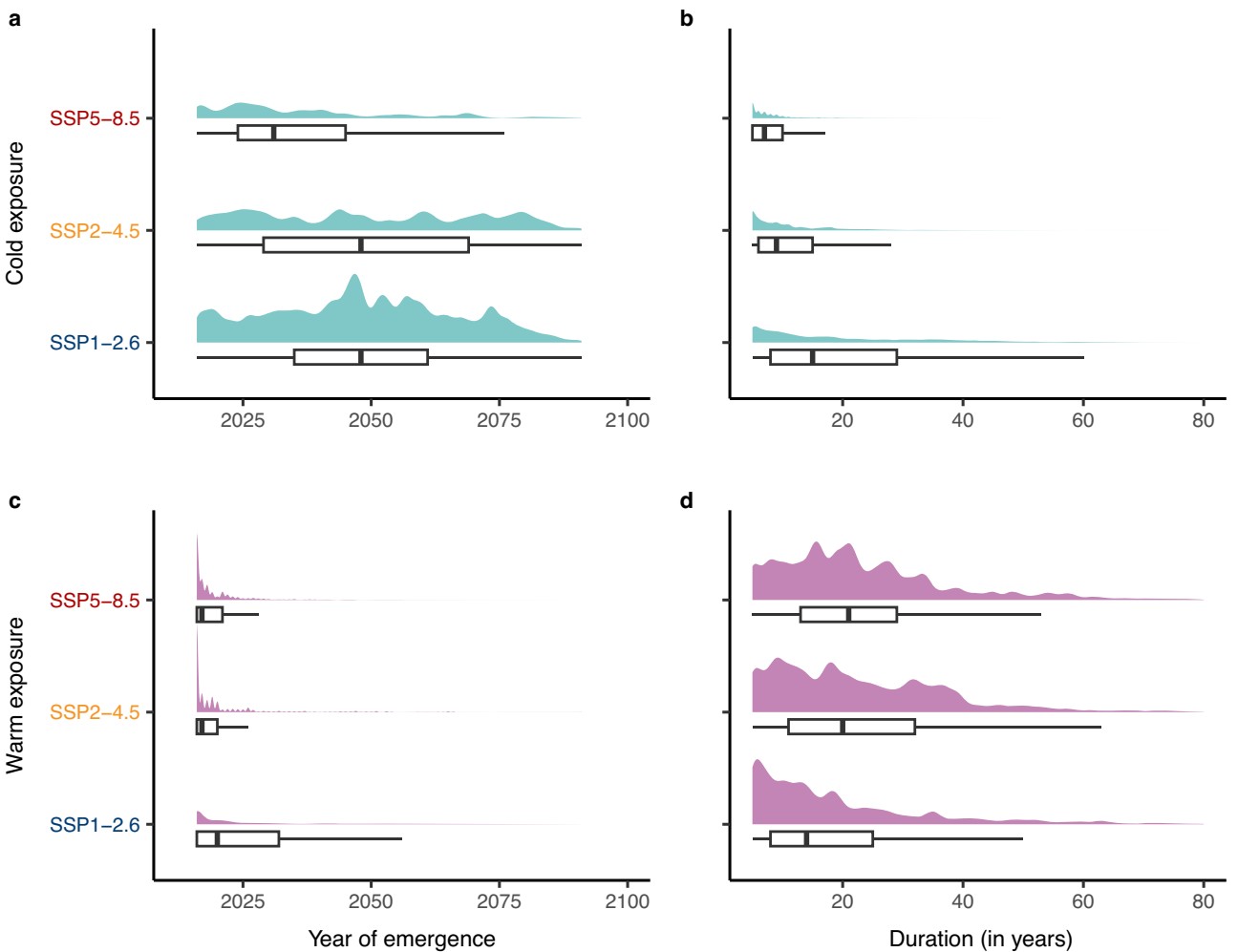

**Fig. 5 | Year of emergence and duration of transient opportunities.** Each figure includes both a density plot scaled proportionally to the number of observations and a boxplot for three emission scenarios. **a** Year of emergence and **b** duration of transient opportunities closed due to exposure to cold temperatures. **c** Year of emergence and **d** duration of transient opportunities closed due to exposure to warm temperatures. Transient opportunities closed due to warm exposure generally arise earlier and last longer than those closed due to cold exposure. All figures show the mean value across nine climate models. Boxplots display the median (centre) and the 25th and 75th percentiles (lower and upper bounds of the box). The upper and lower whiskers extend to ±1.5 times the interquartile range. Outliers beyond the whiskers are not shown. Source Data for this figure can be found in ref. 72.

increases the magnitude of opportunity, but importantly they do not alter the overall trends of timing, abruptness and duration.

Although our findings point to an important role of opportunities in driving marine biodiversity change, several factors might prevent species from taking advantage of opportunities, even if the opportunity is within reach given the dispersal of the species. For instance, biotic interactions such as competition or predation can hinder successful colonisation[43]. Oxygen, pH, and other abiotic variables can also limit the colonisation of newly suitable thermal habitats[32], and ocean currents can function as biogeographic barriers[44], affecting dispersal and constraining the expansion of ranges. In this sense, our analysis might be overestimating the magnitude of opportunities. However, for some species ocean currents and storm-driven events may facilitate access to opportunities beyond the species' dispersal limits[9,45]. Broader thermal niches would increase the number of opportunities projected to arise, enabling species to take advantage of opportunities that were not considered in our analyses.

Despite the evidence of faster range shifts at the poleward edge of species ranges, accelerated range shift rates at the equatorward edge have also been reported[5], as well as a climate-driven decrease in species richness around the equator[25], suggesting that climate change is already causing range contractions. In this sense, it is possible that our

analyses might be conservative and underestimate the magnitude of exposure. Indeed, our analyses do not account for factors such as local adaptation, which could imply narrower thermal limits for some populations and consequently earlier exposure[46]. Furthermore, by considering that exposure happens only after 5 consecutive years of temperatures beyond the realised niche limits of the species, we are not accounting for species that might be at risk after shorter periods of exposure, such as coral reefs[47]. Impacts from disrupted biotic interactions or changes in other important environmental variables (e.g. pH, dissolved oxygen) can also result in the underestimation of exposure. However, species may also have broader thermal niche limits than those estimated here, which could delay or even prevent future exposure. Phenotypic plasticity, evolution, behavioural thermoregulation and acclimation might help species offset future warming stress and avoid local extinction[9,48].

Climate change is expected to affect the vertical structure of ocean temperature[49]. Consequently, opportunities are likely to arise at greater depths, and species might move vertically to avoid exposure near the surface instead of tracking their thermal niches horizontally. Since we focused on sea surface temperature, our study does not account for the temporal dynamics of exposure and opportunity at deeper layers of the ocean, and our findings should not be

extrapolated beyond the epipelagic zone. However, the numerous studies demonstrating that recent changes in sea surface temperature are affecting key ecological attributes of marine organisms suggest that understanding how climate change will affect the thermal seascape at the ocean surface can provide key insights into future climate change risks. Furthermore, if species experience a reduction in their three-dimensional thermal habitat due to the vertical displacement of isotherms and the constraints imposed by their bathymetric depth niches or the photic layer[50], this may drive further horizontal movement. Therefore, understanding the interplay between exposure and opportunities at the surface and greater depths is a critical step towards a more comprehensive assessment of the impacts of climate change in marine ecosystems and an important avenue for future research.

Our study provides insights into the thermal constraints that are likely to drive future marine biodiversity change. We show that the emergence of thermal opportunities is an ongoing process, in which rates will follow a similar pattern until mid-century regardless of the emissions scenarios in the next decade. During this period, exposure events also follow a similar rate across scenarios, but with a much lower magnitude. New opportunities will potentially be the major source of early change in marine temperate and polar ecosystems, even if we manage to rapidly reduce greenhouse gas emissions. Our findings also show that most opportunities are projected to remain open until the end of the century, which increases the chance of successful colonisation if species manage to reach these areas. If emissions remain unabated, then exposure is projected to overtake opportunity as a major source of change towards the end of the century, mostly due to its impacts on tropical ecosystems. Our projections can serve as a valuable early warning system, facilitating targeted monitoring efforts in ecosystems and regions identified by our thermal models as most susceptible to recent and near-term warming. The results of such monitoring can then be used to refine and update short-term projections of climate-driven colonisations and extirpations. These projections will be crucial to adjust conservation and ecosystem management plans for the potential earlier reorganisation of marine assemblages in temperate and polar regions.

## Methods

### Climate data

We estimated exposure and opportunities using sea surface temperature data derived from the Coupled Model Intercomparison Project Phase 6 (CMIP6)[51]. Sea surface temperature is strongly associated with key aspects of marine biodiversity such as species richness[52], abundance[53], phenology[54], and recruitment[55]. Marine organisms are also thermal-range conformers and typically occupy their entire potential latitudinal range, which indicates that their range limits are susceptible to temperature variations[1]. Moreover, studies investigating the impacts of recent climate change on marine species have shown that sea surface temperature is a key driver of distribution shifts[24–28] and community turnover[29]. Therefore, changes in sea surface temperature can be considered a major predictor of future impacts on marine biodiversity, even though it is only a single aspect of climate change.

The selection of models that effectively capture future climate variability is a critical aspect of studies aiming to anticipate the potential impacts of climate change on biodiversity. The recent increase in the number of climate models available has posed a challenge in identifying the optimal models for this purpose. Here, we used phase 3b of the Inter-Sectoral Impact Model Intercomparison Project (ISIMIP3b)[56] to select nine climate models. The model framework of ISIMIP3b selects the best-performing CMIP6 models taking into account process representation, performance in the historical period, data availability, structural independence, and climate sensitivity. Importantly, this set includes models with both low and high climate

sensitivity, providing a good representation of the whole CMIP6 ensemble. The chosen models were CanESM5, CNRM-CM6-1, CNRM-ESM2-1, EC-Earth3, IPSL-CM6A-LR, MIROC6, MPI-ESM1-2-HR, MRI-ESM2-0 and UKESM1-0-LL (Supplementary Table 1). For each model, we downloaded mean monthly estimates from a single projection for the historical run (1850–2014) and for three future (2015–2100) shared socioeconomic pathways: SSP1-2.6 (limiting warming below 2 °C), SSP2-4.5 (mean warming of ~2.7 °C), and SSP5-8.5 (mean warming of >4 °C). We then extracted the climate data to an equal-area 100 km grid and calculated the mean temperature in each grid cell based on the area covered by the climate data. This resolution is recommended for large-scale ecological analyses[57], and its ability to effectively capture risk from climate change has been discussed elsewhere[58]. To obtain yearly estimates, we calculated the mean value for each grid cell using a fixed window between January and December.

### Species data

We obtained species distribution data from AquaMaps[59], the most comprehensive database on marine species distributions. AquaMaps uses an ecological niche modelling approach to estimate species-specific habitat suitability based on environmental variables such as depth, water temperature, salinity, primary productivity, and dissolved oxygen. AquaMaps also incorporates expert knowledge about species occurrences, reducing sampling bias and the chance of species misidentification. To avoid issues from data-scarce species, we only used data from models generated with at least ten occurrence records. Based on depth estimates from AquaMaps, we also excluded from the analyses species that occur exclusively under 200 m depth (epipelagic zone's lower limit), as they are less likely to respond to changes in sea surface temperature. We then transformed the range maps based on the 100 km equal-area grid by overlapping the grid and the AquaMaps range maps. If a species' range intersected with a grid cell, it was considered present in that cell. Our final dataset comprised 21,696 species from 33 phyla. Of these species, 88% are found (yet not exclusively) within the depth range of 0–50 m. Five phyla represent approximately 92% of the species: Chordata (44%, of which 94% are fish), Mollusca (22%), Arthropoda (16%), Cnidaria (6%) and Echinodermata (4%) (Supplementary Fig. 1). Although Chordata represents 44% of the species in our dataset, its median contribution to local species richness (that is, species of a phylum as a percentage of all species in a grid cell) is 81%, followed by Mollusca (8%), Arthropoda (7%) and Echinodermata (3%) (Supplementary Fig. 5).

Instead of presence/absence maps, AquaMaps provides the relative probability of occurrence of each species. We initially used two probability thresholds to define the range of the species: ≥0.5 and >0. While the former is considered standard practice and was adopted in several studies[14,60–63], the latter provides estimates that resemble the extent of occurrence represented by IUCN expert range maps[64]. Since our niche limit estimates are derived from geographic distribution data (see below), the threshold choice could affect our projections. A higher threshold would reduce false positives, provide more conservative range estimates, and presumably narrower thermal limits estimates. A lower threshold would prioritise model sensitivity, detect all known presences, and provide a larger range and wider thermal limit estimates. To evaluate whether AquaMaps and IUCN provide comparable range maps, we obtained the most recent version of IUCN range maps for marine fishes[65], the most species-rich group in our analysis. The IUCN dataset comprises 4189 species of which 2510 are present in the AquaMaps dataset. For each species, we compared the range sizes by dividing the logarithm of the range size obtained from AquaMaps by the logarithm of the range size from IUCN. If the result is greater than 1, the range size from IUCN exceeds that from AquaMaps. Additionally, we compared the overlap between ranges, assessing whether AquaMaps and IUCN estimate the range in the same grid cells. We calculated the overlap by determining the shared grid cells

between the two datasets, then dividing the number of shared grid cells by the total number of unique grid cells from both datasets. Both analyses were conducted using two probabilities of occurrence for AquaMaps (0.0 and 0.5). Results are shown in Supplementary Fig. 11. Range sizes from IUCN were generally smaller compared to AquaMaps 0.0 (median ratio 0.97) and larger compared to AquaMaps 0.5 (median ratio 1.02). The median range overlap between both datasets was >99% for both AquaMaps 0.0 and 0.5. These analyses show that IUCN and AquaMaps provide comparable estimates of species ranges.

To evaluate the impact of threshold selection on our projections, we calculated the cumulative exposure and opportunity (see below) using both probability thresholds (Supplementary Fig. 12) for a 10 km year$^{-1}$ dispersal rate. The results indicated high similarity between the two thresholds, except for exposure under SSP5-8.5, where a probability threshold >0 resulted in estimates 21% higher than a threshold of ≥0.5. This difference can be explained by the fact that a lower threshold results in a larger range estimate, expanding the pool of populations that could potentially be exposed. We, therefore, decided to report only the results obtained using the 0.5 threshold, as this follows standard practice and makes our study comparable to previous ones.

### Thermal niche limits
We estimated realised thermal niche limits using geographical distributions and yearly climate data from the historical period[17,19]. This approach aims to capture the temporal climate variability that is lacking in datasets relying on time-averaged conditions. First, we calculated the maximum ($T_{max}$) and minimum ($T_{min}$) annual mean sea surface temperature experienced by each species across its entire range between 1850 and 2014. We then excluded from calculation values within each cell that lay outside the range of ±3 standard deviations from the mean, to avoid biases in estimates due to climatic outliers or errors in distribution data. After removing outliers from climate data, we calculated the mean $T_{max}$ and $T_{min}$ across the range of each species and also removed the outliers outside ±3 standard deviations. The resulting $T_{max}$ and $T_{min}$ were set as the upper and lower realised niche limits, respectively.

We acknowledge that many species may have shifted their geographical distributions in response to environmental changes during the historical period considered in our analyses, which could impact the estimates of thermal limits. However, many species may also have experienced variability in climatic conditions over space and time, which cannot be fully accounted for when time-averaged climate conditions are used to estimate species' realised niches. Furthermore, our estimates are based only on the climatic conditions where species occur (i.e. realised niche), which are probably narrower than estimates based on their physiological tolerances to climate (i.e. fundamental niche)[66,67]. More importantly, our analyses should not be viewed as an attempt to model future shifts in species distributions. Instead, they serve as an early warning system by predicting where and when species may lose (exposure) or gain (opportunity) thermal suitable habitats. Such information can be used to target monitoring and conservation efforts, as well as provide useful insights for future studies and model predictions. By factoring in historical variability across species ranges, we believe our estimates provide a more meaningful approximation of species thermal tolerances than if only current, averaged climate estimates had been used.

### Constraints to thermal opportunities
We used depth and dispersal to constrain the cells where opportunities could emerge. We first estimated the depth values within each grid cell using data from the ETOPO1 Global Relief Model[68,69]. We then used species maximum and minimum depth intervals from AquaMaps[59] to classify the cells. Only grid cells in which the depth range of the species fully or partially overlapped the depth range of the cell were considered suitable for opportunity emergence. For species flagged as

pelagic by AquaMaps, we also considered suitable cells in which the depth range of the species was higher than the depth range of the grid cell, as these species are less likely to respond to variations in bottom depth.

Several studies have assessed the speed of recent range shifts in range boundaries of marine organisms, with average estimates ranging from 2.4 ± 8.7 km year$^{-1}$ [30] to 7.20 ± 1.35 km year$^{-1}$ [6]. However, some species (e.g. some Actinopterygii fishes) have shifted their leading edges at rates of 50 km year$^{-1}$ or more[5,31]. To incorporate dispersal ability in our framework we constrained the grid cells in which opportunities could arise by creating a buffer around the distribution of each species. We used two buffer sizes based on the average and the maximum dispersal rates of marine organisms (i.e. 10 km year$^{-1}$ and 50 km year$^{-1}$). Our buffer size was calculated by multiplying these rates by the number of years in our future climate models (86 years), generating a final buffer size of 860 and 4300 km. Only grid cells within the buffer and outside the current distribution range of the species were considered suitable for opportunity emergence. Opportunities could arise anywhere within the buffer, as long as the depth is suitable, meaning that they could arise in regions not necessarily adjacent to the species' range. Opportunities could also arise for grid cells within the current range of the species, but only if an exposure event has occurred previously in the grid cells. Maps illustrating workflow used for constraining and estimating opportunities are presented in Supplementary Fig. 13.

Although marine ectotherms tend to fully occupy their latitudinal ranges, the same does not occur with their longitudinal ranges[1]. Consequently, many grid cells where a species is not predicted to occur could have sea surface temperatures within the species estimated thermal limits (niche unfilling). The potential absence of the species in those cells can have several causes, such as the unsuitability of environmental parameters other than temperature (e.g. pH, salinity, dissolved oxygen), or the absence of favourable biotic interactions. To account for this, we excluded grid cells in which temperature falls within the thermal limits of a species, but are classified as unsuitable by AquaMaps.

### Thermal exposure and opportunity
We estimated the timing of future thermal exposure following the biodiversity climate horizon framework[17]. This framework uses species thermal limits estimates and yearly climate change projections to forecast the temporal dynamics of species exposure to climate change. Specifically, we estimated the number of years each species in each grid cell would experience temperatures beyond their realised thermal niche limits (i.e. >$T_{max}$ or <$T_{min}$). We then classified as exposed species projected to experience at least five consecutive years of unprecedented temperatures, defining the time of exposure as the first year of the series. Earlier tests demonstrated that increasing the number of consecutive years used (e.g. 20 years) had little impact on exposure patterns[17]. Exposure was estimated only across the native range of the species.

We used the same rationale to estimate the timing of future thermal opportunities. However, rather than estimating the number of years that species would experience temperatures beyond their thermal niche limits, we estimated the number of years grid cells classified as suitable for opportunity emergence would experience temperatures within the thermal niche limits of each species (i.e. ≤$T_{max}$ and ≥$T_{min}$). An opportunity emerges whenever the grid cell experiences five or more consecutive years of temperature within limits. The timing of opportunity is also defined as the first year of the series. To summarise the temporal dynamics of exposure and opportunity estimates, we used the following metrics established by ref. 17: magnitude of exposure, which is the maximum number of species exposed in a grid cell; magnitude of opportunity, the maximum number of opportunities that arose locally (calculated as a

percentage of the local species richness); timing of exposure, the median year in which exposure occurred; timing of opportunity, the median year in which opportunity arises in a grid cell; abruptness of exposure, the percentage of species exposed in the decade of maximum exposure; abruptness of opportunity, the percentage of opportunities within the decade where the maximum number of opportunities arose.

To further explore the temporal dynamics of opportunities, we analysed their duration through the century. First, we assessed whether or not opportunities would remain open until the end of the century. Based on that, we separate them into two groups: persistent and transient. Opportunities that were still open at 2100 were classified as persistent. If, after the opportunity emergence, the grid cell experiences five or more consecutive years of temperatures beyond the niche limits of the species for which the opportunity is being calculated, we consider that this opportunity closed, and classified it as transient. To understand why opportunities close, we further divided transient opportunities into two groups: cold- and warm-exposed. Cold-exposed opportunities were defined as those that experienced five consecutive years of temperatures below the thermal limits of the species for which the opportunity is being calculated, while warm-exposed opportunities are those that have experienced five consecutive years of temperatures above those limits. Theoretically, opportunities could emerge and close several times in the same cell. All these events are counted in our analyses.

Determining the number of years a region must remain thermally suitable for successful colonisation is challenging, as species responses are highly individualistic and can be influenced by factors other than climate (e.g. dispersal ability, competition, and diet)[8,9]. Although marine species, in general, are closely tracking temperature changes, many species still lag behind climate change. For species with high mobility and short generation times, a single year of suitable temperatures may be sufficient for range expansion[70], especially if the opportunity arises close to the existing range of the species. In contrast, for species with lower mobility or narrower dietary breadth[9], there may be a delay between thermal opportunities arising and colonisation occurring. For such species, more consecutive years of suitable temperature might be needed for successful colonisation. Therefore, based on the biology of range dynamics we selected five consecutive years as an interval to capture the dynamics of thermal opportunity for species that may shift their distribution quickly and those that may lag behind climate change. Nevertheless, we emphasise again that our analyses are not designed to predict range shifts, but rather to provide an overview of temporal dynamics of habitat gains and losses in the thermal seascape. Lastly, since most thermal opportunities identified in our analysis are projected to remain open until the end of the century (Fig. 4), increasing the number of consecutive years used to define a thermal opportunity beyond five years would have little influence on the spatio-temporal patterns observed. Increasing the number of consecutive years could, however, obscure information on transient thermal opportunities as the number of transient opportunities is low when compared to non-transient opportunities. All analyses and visualisations were done in R[71] version 4.3.3.

### Reporting summary
Further information on research design is available in the Nature Portfolio Reporting Summary linked to this article.

## Data availability
No primary data were generated in this study. CMIP6 pre-processed climate data used in the analysis were downloaded from https://esgf-index1.ceda.ac.uk/search/cmip6-ceda/. CMIP6 data references are provided in Supplementary Table 1. Pre-processed species data are available from AquaMaps[59] (https://aquamaps.org/). Species distribution maps presented in this study have been included with permission from the AquaMaps team. Sample post-processed climate data, species data, and Source data for Figs. 1–5 are provided in ref. 72.

## Code availability
The R scripts required to reproduce the analyses and recreate the figures are provided in ref. 72.

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

## Acknowledgements

This work was supported by the National Socio-Environmental Synthesis Center (SESYNC) with funding from the National Science Foundation DBI-1639145 (C.H.T.), the FLAIR Fellowship Programme—a partnership between the African Academy of Sciences and the Royal Society funded by the UK Government's Global Challenges Research Fund (C.H.T., A.L.P., and A.S.M.), the European Union's Horizon Europe Research and Innovation Programme through the RESCUE project no. 101056939 (C.H.T. and A.S.M.), and Schmidt Sciences (C.H.T.). We thank Kerry-Anne Grey for her helpful comments on the manuscript.

## Author contributions

A.S.M., A.L.P., C.M. and C.H.T. conceived the ideas and designed the study. C.H.T. and A.L.P. acquired funding for the project. K.K., C.G. and K.K.-R. provided data and conceptual advice. A.S.M. performed the analyses and wrote the manuscript, with significant contributions from C.H.T. and A.L.P. All authors revised the manuscript and gave final approval for publication.

## Competing interests

The authors declare no competing interests.
