## [Peer Review File · Nature Communications]

Temporal dynamics of climate change exposure and opportunities for global marine biodiversityREVIEWER COMMENTS

Reviewer #1 (Remarks to the Author):

The manuscript titled 'Temporal Dynamics of Future Thermal Exposure and Opportunities for Marine Biodiversity' by Meyer et al. describes analyses of various climate change metrics (exposure, opportunity, abruptness, timing) for approximately 22,000 species derived from AquaMaps across three climate change scenarios (SSP1-2.6, SSP2-4.5, SSP5-8.5), using sea surface temperature as a proxy. The most important findings are that “opportunities” are projected to arise earlier this century and that are projected to arise at similar rates regardless of emissions pathway. They also found that "exposure" will become stronger only at the end of the century, and opportunities will be more predominant throughout the entire period.

Clarity and context:

The abstract was succinct and provided an accessible overview of the study, while the introduction offered a thorough background on relevant literature. Overall, the manuscript is clear and well-written.

Originality and significance:

In my opinion, using sea surface temperature as a proxy for communities may no longer be the most suitable approach for global studies, and it may not have received the attention it truly needs (for context, please consider the following references: Jorda et al. 2020, Brito-Morales et al. 2020, Santana-Falcon et al. 2022). For example, the high seas (areas beyond EEZ) extend to a depth of approximately 8 km as some of the AquaMaps species used in this study do. Is it appropriate to relate what happens with the temperature in the first 5 m of the ocean to what would happen at the mesopelagic layer, for example (200-1000 km)? The answer is no, and the citations mentioned above clearly indicate that might not be suitable. Although the authors have referenced why sea surface temperature might be appropriate, it's worth noting that the studies are before 2020.

Another important fact is that in the ocean species filled their thermal niches not only in the horizontal but also in the vertical, and there's scientific evidence of that. Implying that

species moves only in the horizontal is a big assumption and a simplistic way of looking the impacts of climate warming in the ocean.

Lastly, I have genuine concerns about the novelty of this work. While the authors have done a good job highlighting global patterns of exposure, opportunities, abruptness, and timing, I see significant similarities to the findings of Trisos et al. 2019 (<https://doi.org/10.1038/s41586-020-2189-9>). The patterns and figures showed in this paper closely resemble what has already been described (just one example is Fig1 - same pattern as Fig1a of Trisos et al. 2019; same with Fig3).

Suggested improvements:

54-56: Is this statement true only for the epipelagic layer?

62: The statement "marine organisms have been found to be particularly sensitive to warming" requires a citation.

111: Using yearly sea surface temperature data is problematic in global analyses. Please refer to Jorda, Brito-Morales, and Santana-Falcon for more information.

141-156: The entire paragraph is based on SSP585, but the analysis was conducted using three different climate scenarios. Is there a specific reason for focusing on only one scenario? It would be more informative to compare the findings across all three scenarios. Perhaps a new Figure 1, incorporating some Extended Data, would be appropriate here.

150-151: Please clarify how many communities were analysed. Are these communities defined by the number of species per pixel? If so, please reiterate this in the main text as "communities analysed (i.e., species per pixel)."

168-169: "Risks of thermal exposure decrease substantially under intermediate (SSP2-4.5) and low emission (SSP1-2.6)" - Some studies have suggested that SSP126 would be beneficial only for species living in the epipelagic layer and may not apply to those in the mesopelagic layer. This supports the message regarding the use of SST as a proxy for the entire water column (see Brito-Morales et al. 2020 and Santana-Falcon et al. 2022).

175-177: "These results suggest that reducing greenhouse gas emissions is expected to have a greater impact on exposure than on opportunities." This may be true primarily for the surface of the ocean.

192-196: This result is interesting but also somewhat confusing. It cannot be simplified in

such a manner. The authors should provide an explanation for this claim.

265-268: Again, this could only be true for the epipelagic layer as the deep ocean is likely to continually warm, even under a SSP1-2.6

369-385: Section quite confusing. It's unclear why the authors have chosen to address dispersal rates here instead of incorporating them into the main framework. This content should ideally be integrated into the entire manuscript rather than placed at the end.

References:

Brito-Morales, I. et al. Climate Velocity Can Inform Conservation in a Warming World. *Trends Ecol. Evol.* 33, 441–457 (2018).

Jorda, G. et al. Ocean warming compresses the three-dimensional habitat of marine life. *Nat. Ecol. Evol.* 4, 109–114 (2020).

Santana-Falcón, Y. & Séférian, R. Climate change impacts the vertical structure of marine ecosystem thermal ranges. *Nat. Clim. Change* 1–8 (2022)

Reviewer #2 (Remarks to the Author):

I really enjoyed reading this paper and I think there is beauty and power in the simplicity and straight-forwardness of the approaches the authors used. I think this is an important and insightful study that will make a good fit for Nature Communications. The quality of the manuscript is excellent, and it is clearly very well polished and detailed. Thus, my overall impression of the study is very favorable. That said, I do see one major issue. This is a study on biodiversity, yet the biological aspect is lacking and mostly ignored in the paper. All that is said regarding the biota is that there were 21,696 species from 33 phyla. The authors do not state, nor consider (as far as I can tell), the representation by these different taxa. For instance, lumping all biodiversity together and making one general prediction seems appropriate when you have equal representation by all taxa. However, if for a given grid cell, 90% of your biodiversity is fishes and the rest is spread among other organisms, are your results truly representative of how marine biodiversity will change in the future? Moreover, if the proportion of different taxa or phyla present within each grid cell is highly variable across space (in some instances due to sampling effort by data availability), won't this lead to some biases and non-comparable trends? For instance, wealthy countries have

much greater capacity for biodiversity monitoring, particularly bottom-trawl surveys. So, could you end up with a greater proportion of fishes in your grid cells on coasts near wealthy countries than in developing countries or some other potential biases? I think this needs to be considered and minimally discussed in the paper. I also think the authors should provide information on how these ~22,000 species are distributed across major taxonomic groups and whether this is consistent across space. Another thing to consider along these lines, is whether there is a high degree of disparity in responses among taxa – are most of the exposures or opportunities are due to a single taxonomic group within a given location, or are all taxa having similar responses?

The other issue I see related to the biological aspect is the maximum dispersal distance. Attributing a single value to all taxa seems problematic. I appreciate that the authors tested both 10 and 50 km/year. I think 10 is much more realistic and that 50 is likely a large overestimate, but it's good to see congruency between both values. However, if 50 km/year is near the maximum value observed for the most mobile taxa, I would suspect this leads to a strong over-estimation. For instance, if the database contains many sessile organisms with limited larval dispersal (e.g., sponges), those taxa are certainly not shifting at 50 km/year. Would it be possible to calculate unique dispersal values for different taxa? Why not allow sponges, fishes, seaweeds, etc. to all have their own dispersal abilities, this would lead to a much more realistic projection.

Those are my major comments. I would like to see the reviewers consider these thoughts and minimally i) provide information on the representation of different taxa in the data, and ii) discuss these implications of their paper.

Beyond that I have no major recommendations as I find the paper is exceptional in all aspects. Minor and line-by-line comments below:

Abstract – if you have room, I would recommend defining exposure and opportunities in the abstract

Abstract – I think you could be more precise here – there are many studies on temporal

biodiversity projections, but most are for a single time period, so annually-resolved or continuous assessments are what is lacking.

Line 66 – hourly temperatures? Is this really defined on an hourly time scale?

Line 154: basing this statistic on 5% of species richness seems a bit low, no? 10% or higher would be more powerful? I think it's reasonable to expect a change in 5% of current richness almost everywhere, so it is not a very "big" or surprising result?

Figure 2 – a dashed or solid black horizontal line at 0 could help emphasize.

Discussion – I think you should speak about other potential limiting factors like competition, habitat suitability, or other biotic and abiotic conditions that might limit species from taking advantage of opportunities that arise.

Reviewed by:

Matthew McLean

[editorial note: contact details redacted]

Reviewer #3 (Remarks to the Author):

This manuscript calculates two things for the global oceans: first, how many of the species currently modeled to occupy any given grid cell will have their thermal niches exceeded under different climate futures (and when); and second, when and where thermally suitable climates will arise for the same species in adjacent locations. There are two things I especially like about this paper. The first is that it quantifies possible habitat losses on the same footing as possible habitat gains, which is a bit tricky to do—colonization processes happen very differently from extinction processes; the authors circumvent this difficulty by focusing just on thermal habitat, not actual species distributions. Relatedly, the second thing I want to compliment is the careful way the authors present their analysis and results in the context of changing thermal habitat suitability rather than projected changing species distributions. This is a chronic issue in global change biology, when researchers conflate a

changing climate with changing biogeography without first testing that hypothesis; I really appreciate how this paper emphasizes that thermal exposure and opportunity set some bounds on possible future species shifts but do not in themselves represent biodiversity projections.

I should note that it is impossible to fully review this paper because it does not comply with the Nature Communications policy that peer review of code is required when the code is central to manuscript results, as in this case (see <https://www.nature.com/nature-portfolio/editorial-policies/reporting-standards#availability-of-computer-code>). The manuscript is also missing statements on Data Availability and Code Availability. Thus, the Reporting Summary for this submission should be viewed as incomplete. That said, I have provided as complete a review as I can of the manuscript alone.

The methods are relatively straightforward and well-described. They also build on existing studies, including some by the authors themselves. Indeed, given that the entire “exposure” half of the analysis appears to be virtually identical to Pigot et al. 2023, I was surprised to not see that manuscript cited and discussed here. I imagine this is partly due to the journal’s constraint on number of citations; the authors should feel free to add a reference to and discussion of Pigot et al. 2023, and blame this Reviewer if it puts them over the recommended number of references!

Like many analyses of global marine biogeography, this project used Aquamaps. Aquamaps is both the most convenient tool for this purpose and highly limited in its accuracy and applicability. The authors may be aware of other options for mitigating this, but my understanding based on O’Hara et al. 2017 is that best practices include testing how sensitive the results are of studies using Aquamaps by (1) using IUCN range maps for comparison and/or (2) subsetting Aquamaps to only the species whose range maps are expert-reviewed. Since the authors of the present study had fairly relaxed criteria for which taxa to include (only 10 occurrence points used to model the entire range), I think an analysis of the sensitivity of these results to Aquamaps’ often-erroneous range estimation algorithm would be appropriate. (Another issue: The authors acknowledge this when discussing the different probability of occurrence thresholds, e.g. in Extended Data Fig 5, but

a different point I should raise about Aquamaps is that it models marine species' ranges as being absolutely huge; this is a problem when Aquamaps is used in studies like this one because it models vast swaths of the oceans as being part of any given species' range when that species has never been found there. The authors ratcheted down the threshold for probability of occurrence in their analysis, from 0.5 to 0. I probably would have ratcheted it up to a much higher value to make it more likely that the modeled ranges are reflective of where species can actually be found.)

One technical question: Am I correct in interpreting L546-548 to mean that from one year to the next, thermal opportunity was calculated for every species within an area representing how far they could be expected to disperse (based on average or maximum dispersal distances) in 85 years? If the focus of this manuscript is the temporal dynamics of changing climates, it seems puzzling to assume each species' has access to 85 years' worth of dispersal distance in each year. I understand long-distance dispersal is a factor but this is not a particularly direct way to quantify its potential impacts; indeed these distances are so large even the smaller value (850 km), if considered an upper limit for a single year, essentially represents no barriers to dispersal. It might be more realistic to re-run the annual projections giving each species the authors' own estimates of dispersal distance per year (10-50 km) to see how it would constrain the projections.

I commend the authors for analyzing multiple climate scenarios, which numerous recent articles have argued is key for maximizing applicability of climate and biodiversity research to e.g. the IPCC process (Schoeman et al. 2023). However, the authors offer no explanation for the choice to present main text results based on SSP5-8.5, the high-emissions, no-mitigation scenario that is generally considered to be an unlikely future (Hausfather & Peters 2020). Actually, unless I missed it, the choice of scenarios and what each mean for the future is not really introduced anywhere in this manuscript. I'd encourage the authors to add a few sentences to the Introduction along these lines, because in any manuscript talking about the future, it's important to introduce which potential future we're talking about and why. The manuscript text implies that the main results analyzed a likely future because it uses the word "will" frequently in reference to the modeled projections. Given that, I suggest that the authors re-center their results around SSP2-4.5 (currently considered the

most probable of their three scenarios), or re-frame the paper to explain why they aimed to illustrate a worst-case scenario for the oceans that is thankfully unlikely to come to pass. Presenting all three scenarios in-text, which the authors do for some results (e.g., Fig. 2), is another option. Guidance for choosing and discussing climate scenarios that may be helpful can be found in Burgess et al. 2023.

L562: This sentence doesn't make sense to me (it seems to say "when temperature becomes unsuitable, we classify that cell as suitable in the future"). Is "outside" meant to be "within"? (It also doesn't seem to follow from the previous sentence, which I interpret to say "when our model said temperature was suitable but Aquamaps said the cell was uninhabitable, we believed Aquamaps because it uses other non-temperature parameters in its model.")

Burgess, M.G., Becker, S.L., Langendorf, R.E., Fredston, A. & Brooks, C.M. (2023). Climate change scenarios in fisheries and aquatic conservation research. *ICES J. Mar. Sci.*, 80, 1163–1178.

Hausfather, Z. & Peters, G.P. (2020). Emissions – the 'business as usual' story is misleading. *Nature*, 577, 618–620.

O'Hara, C.C., Afflerbach, J.C., Scarborough, C., Kaschner, K. & Halpern, B.S. (2017). Aligning marine species range data to better serve science and conservation. *PLOS ONE*, 12, e0175739.

Pigot, A.L., Merow, C., Wilson, A. & Trisos, C.H. (2023). Abrupt expansion of climate change risks for species globally. *Nat. Ecol. Evol.*, 1–12.

Schoeman, D.S., Gupta, A.S., Harrison, C.S., Everett, J.D., Brito-Morales, I., Hannah, L., et al. (2023). Demystifying global climate models for use in the life sciences. *Trends Ecol. Evol.*, 38, 843–858.

Comments to Reviewers

Firstly, we would like to thank the reviewers for their time and effort on behalf of our manuscript. We are sincerely grateful for their insightful review.

Reviewer #1:

The manuscript titled 'Temporal Dynamics of Future Thermal Exposure and Opportunities for Marine Biodiversity' by Meyer et al. describes analyses of various climate change metrics (exposure, opportunity, abruptness, timing) for approximately 22,000 species derived from AquaMaps across three climate change scenarios (SSP1-2.6, SSP2-4.5, SSP5-8.5), using sea surface temperature as a proxy. The most important findings are that "opportunities" are projected to arise earlier this century and that are projected to arise at similar rates regardless of emissions pathway. They also found that "exposure" will become stronger only at the end of the century, and opportunities will be more predominant throughout the entire period.

Clarity and context:

The abstract was succinct and provided an accessible overview of the study, while the introduction offered a thorough background on relevant literature. Overall, the manuscript is clear and well-written.

A1: We thank the Reviewer for the positive feedback.

Originality and significance:

In my opinion, using sea surface temperature as a proxy for communities may no longer be the most suitable approach for global studies, and it may not have received the attention it truly needs (for context, please consider the following references: Jorda et al. 2020, Brito-Morales et al. 2020, Santana-Falcon et al. 2022). For example, the high seas (areas beyond EEZ) extend to a depth of approximately 8 km as some of the AquaMaps species used in this study do. Is it appropriate to relate what happens with the temperature in the first 5 m of the ocean to what would happen at the mesopelagic layer, for example (200-1000 km)? The answer is no, and the citations mentioned above

clearly indicate that might not be suitable. Although the authors have referenced why sea surface temperature might be appropriate, it's worth noting that the studies are before 2020.

Another important fact is that in the ocean species filled their thermal niches not only in the horizontal but also in the vertical, and there's scientific evidence of that. Implying that species moves only in the horizontal is a big assumption and a simplistic way of looking the impacts of climate warming in the ocean.

A2: We agree with the Reviewer that looking into greater ocean depths would provide a more comprehensive picture of the projected impacts of climate change in the ocean. Opportunities can emerge at deeper layers of the ocean, and species might move vertically to track their thermal niches such that a vertical component may add additional insight. Few studies to date have done so, and this represents a crucial gap for future research. However, we believe this does not render SST unsuitable for studies like ours, or that the projected changes in the horizontal dimension are incorrect or uninformative as an important aspect of future changes in marine biodiversity globally. SST is widely recognized as a major component of marine species' thermal niche. This notion is supported by the strong relationship between SST and key ecological attributes including species richness and range margins, and also by a large body of evidence showing that marine organisms are changing their distributions at rates and in directions that can be explained by changes in SST, as highlighted in the Methods (lines 489-493). Studies published after 2020 further reinforce this perspective and we now cite these in the manuscript (e.g. Chaudhary et al. 2021, Oke et al. 2022, Rutterford et al. 2023)¹. Although no single variable can provide a

¹References:

- Chaudhary et al. (2021) Global warming is causing a more pronounced dip in marine species richness around the equator. PNAS 118, e2015094118.
- Oke et al. (2022) Sea-surface temperature anomalies mediate changes in fish richness and abundance in Atlantic and Gulf of Mexico estuaries. Journal of Biogeography 49, 1609-1617.
- Rutterford et al. (2023) Sea temperature is the primary driver of recent and predicted fish community structure across Northeast Atlantic shelf seas. Global Change Biology 29, 2510-2521.

complete picture of the impacts of climate change on marine ecosystems, the literature shows that SST is an important predictor that can provide valuable insights into current impacts and potential future impacts.

Thus, we agree with the reviewer that the influence of temperature changes at greater depths is worthy of study, but is beyond the scope of this manuscript as moving from 2D to 3D climate modelling of exposure and opportunities would require a complete redevelopment of the entire analysis, as well as extensive sensitivity analyses to assess how different vertical migration rates would affect the results, and furthermore the use of SST only still provides valuable insights into the dynamics of climate change for marine biodiversity. We now acknowledge clearly in the Introduction (lines 108-115) that our aim and our findings are applicable to the uppermost layer of the ocean and do not capture dynamics at greater depths, and note that 88% of the species in our dataset are found (yet not exclusively) within the depth range of 0-50m – this information is also now included in the Methods (lines 529-530). The depth range in our study is a crucial range where variations in SST are likely to have a strong influence. Even though these species might move vertically to escape exposure, they are still exposed near the surface.

In order to further incorporate the points raised by the Reviewer, we added a new paragraph in the Discussion (lines 444-460). In this paragraph, we further discuss the limitations of our study regarding the use of SST and the implications of not considering vertical migration. We also state that our findings should not be extrapolated beyond the epipelagic zone, and underscore the importance of including vertical movements into future research.

Finally, we would like to emphasize that we excluded species occurring exclusively below 200m from our dataset (refer to the Methods section, lines 523-526). We made this exclusion because we recognise that the focus of our study is the epipelagic zone. It was not our goal to present our findings as if they could be extrapolated to deeper layers of the ocean. We acknowledge that our wording throughout the manuscript might not have been sufficiently clear in this

regard. This is now clarified in the Introduction (lines 112-113) and reinforced in other parts of the text when describing the results (see lines 159, 213, 290, and 453).

Lastly, I have genuine concerns about the novelty of this work. While the authors have done a good job highlighting global patterns of exposure, opportunities, abruptness, and timing, I see significant similarities to the findings of Trisos et al. 2019 (<https://doi.org/10.1038/s41586-020-2189-9>). The patterns and figures showed in this paper closely resemble what has already been described (just one example is Fig1 - same pattern as Fig1a of Trisos et al. 2019; same with Fig3).

A3: This work makes novel advances and presents several new methods and findings not found in the Trisos et al. 2019 paper. Specifically, this new work is an important step forward in quantifying potential habitat change by looking both at thermal opportunity and exposure in a consistent framework. Trisos et al. looked only at exposure and did not consider any analysis of thermal opportunities. In contrast, in this manuscript we provide a novel and comprehensive analysis of how climate-driven changes in both thermal opportunities and exposure, as well as their temporal dynamics, and their interplay, could shape marine biodiversity up to 2100. Our new analysis in this paper provides major new findings that are not found in the Trisos et al. 2019 work. First, we show that thermal opportunities arise earlier and more gradually when compared to thermal exposure (Figs. 1, 2 and 3). Second, we show that thermal opportunities will generally remain open until the end of the century (Fig. 4). Third, we quantify and map where transient opportunities are projected to arise (Fig. 4) and for how long they are projected to remain open (Fig. 5). Finally, we quantify and map the causes of transient opportunities (that is, warm vs cold exposure) (Fig. 5). Taken together these novel findings integrate analyses of thermal exposure and opportunity and represent a major advance over the work of Trisos et al. 2019 in terms of understanding the temporal dynamics of both potential habitat losses and gains in a single framework.

In addition, while it is true that we find similar exposure patterns to Trisos et al. 2019 these are simply presented as context for understanding the role of exposure in relation to opportunities and the interplay of the two, since the analysis of opportunities and their interplay with exposure is a major focus of this paper. For example, Figure 1 in our manuscript is substantially novel compared to Trisos et al. 2019 because it presents an integrated picture of both exposure and opportunities across the global oceans. The same is true of Figure 3.

Suggested improvements:

54-56: Is this statement true only for the epipelagic layer?

A4: This statement referred to the overall redistribution of global biodiversity. However, in order to keep within the word count while responding to reviewer comments, we have removed the paragraph containing this sentence in the current version of the manuscript.

62: The statement "marine organisms have been found to be particularly sensitive to warming" requires a citation.

A5: Citation added.

111: Using yearly sea surface temperature data is problematic in global analyses. Please refer to Jorda, Brito-Morales, and Santana-Falcon for more information.

A6: We agree SST does not present a complete picture of potential future climate change dynamics in the ocean. We have responded to this above (response A2) and we have added additional clarification and discussion throughout the manuscript on the scope of our study and the importance and potential limitations of using SST (see Introduction lines 108-115, Discussion lines 444-460).

141-156: The entire paragraph is based on SSP585, but the analysis was conducted using three different climate scenarios. Is there a specific reason for focusing on only one scenario? It would be more informative to compare the findings across all three scenarios. Perhaps a new Figure 1, incorporating some Extended Data, would be appropriate here.

A7: We agree with the Reviewer that including the findings of all three scenarios is important. We have now incorporated the results from the remaining SSPs not only in Fig. 1 but also in Fig. 3 and Fig. 4. All figures of the manuscript now show the results from all three SSPs.

150-151: Please clarify how many communities were analysed. Are these communities defined by the number of species per pixel? If so, please reiterate this in the main text as "communities analysed (i.e., species per pixel)."

*A8: We now use the word assemblages throughout the manuscript. We added defined assemblage as "*species in a 100 km grid cells*" (line 124) and also stated the number of assemblages analyses (N = 41,220) on lines 123-124.*

168-169: "Risks of thermal exposure decrease substantially under intermediate (SSP2-4.5) and low emission (SSP1-2.6)" - Some studies have suggested that SSP126 would be beneficial only for species living in the epipelagic layer and may not apply to those in the mesopelagic layer. This supports the message regarding the use of SST as a proxy for the entire water column (see Brito-Morales et al. 2020 and Santana-Falcon et al. 2022).

A9: We removed this sentence from the manuscript. However, have revised text to be clear that our results have limits for greater depths (see response A2).

175-177: "These results suggest that reducing greenhouse gas emissions is expected to have a greater impact on exposure than on opportunities." This may be true primarily for the surface of the ocean.

A10: We have revised this sentence to read “Reducing future greenhouse gas emissions has a greater impact on reducing thermal exposure than opportunities in the upper layer of the ocean” (lines 158-159).

192-196: This result is interesting but also somewhat confusing. It cannot be simplified in such a manner. The authors should provide an explanation for this claim.

A11: We meant that the high number of thermal opportunities projected to emerge at higher latitudes suggests that the temperature in these regions will likely become suitable for many species from lower latitudes. Consequently, the significance of other abiotic factors (e.g. oxygen and pH) in constraining or promoting range shifts becomes more relevant for those species. We have reworded the text to articulate our point more clearly: "The early emergence of thermal opportunities across all scenarios also suggests that for many species temperature might not be the main abiotic constraint for poleward range shifts in the near future. Consequently, the role of other abiotic factors such as oxygen and pH in constraining poleward range shifts may become more relevant in the next two decades^{32,33}. Understanding how these factors will change in the future is likely to improve projections of near to mid-term range shifts in the epipelagic layer of the ocean." (lines 207-213).

265-268: Again, this could only be true for the epipelagic layer as the deep ocean is likely to continually warm, even under a SSP1-2.6

A12: We have added “in the epipelagic layer” to this sentence (line 290).

369-385: Section quite confusing. It's unclear why the authors have chosen to address dispersal rates here instead of incorporating them into the main framework. This content should ideally be integrated into the entire manuscript rather than placed at the end.

A13: We now mention in the Introduction the two dispersal rates used to constrain opportunities (lines 136-140). In response to the comments of the other reviewers, we changed the main focus of the manuscript to the results obtained using a 10 km year⁻¹ dispersal rate and now also refer to the results from a 50 km year⁻¹ dispersal rate throughout the manuscript. As suggested by the Reviewer, we have also moved the dispersal results that were in this paragraph earlier in the manuscript into the relevant sections on magnitude (lines 166-167 and lines 179-181), timing and abruptness (lines 282-283), and persistent/transient opportunities (lines 308-310).

Dr Matthew McLean (Reviewer #2):

I really enjoyed reading this paper and I think there is beauty and power in the simplicity and straight-forwardness of the approaches the authors used. I think this is an important and insightful study that will make a good fit for Nature Communications. The quality of the manuscript is excellent, and it is clearly very well polished and detailed. Thus, my overall impression of the study is very favorable.

A14: We thank Dr McLean for the positive feedback.

That said, I do see one major issue. This is a study on biodiversity, yet the biological aspect is lacking and mostly ignored in the paper. All that is said regarding the biota is that there were 21,696 species from 33 phyla. The authors do not state, nor consider (as far as I can tell), the representation by these different taxa. For instance, lumping all biodiversity together and making one general prediction seems appropriate when you have equal representation by all taxa. However, if for a given grid cell, 90% of your biodiversity is fishes and the rest is spread among other organisms, are your results truly representative of how marine biodiversity will change in the future? Moreover, if the proportion of different taxa or phyla present within each grid cell is highly variable across space (in some instances due to sampling effort by data availability), won't this lead to some biases and non-comparable trends? For instance, wealthy countries have much greater capacity for biodiversity monitoring, particularly bottom-trawl surveys. So, could you end up with a greater proportion of fishes in your grid cells on coasts near wealthy countries than in developing countries or some other potential biases? I think this needs to be considered and minimally discussed in the paper. I also think the authors should provide information on how these ~22,000 species are distributed across major taxonomic groups and whether this is consistent across space. Another thing to consider along these lines, is whether there is a high degree of disparity in responses among taxa – are most of the exposures or opportunities are due to a single taxonomic group within a given location, or are all taxa having similar responses?

A15: Dr McLean raised several valid points here. Sampling bias is pervasive in biodiversity datasets. In order to mitigate its impact on the data, Aquamaps

combines species distribution modelling, expert knowledge, and information on habitat usage to estimate species ranges. In doing so, Aquamaps represents an improvement compared to 'expert opinion' distribution maps. Nevertheless, sampling bias cannot be entirely eliminated from the dataset, and assessing the extent to which our results are affected by it is not trivial, as information on sampling efforts is lacking. However, we quantified the contribution of the phyla with most species in our dataset to our results and incorporated this information in the paper, as suggested by Dr McLean. In the introduction, we now write “A total of 21,696 species were used with five phyla contributing 92% of the species: chordates (44%, mainly fish), molluscs (22%), arthropods (16%), cnidarians (6%), and echinoderms (4%)” (lines 95-97). For these five phyla, species richness is greatest along coastlines, especially in the tropics, with Chordata being the most species-rich group in the open oceans (Extended Data Fig. 1). We also include further information in the Methods line (lines 532-536) “*Although Chordata represents 44% of the species in our dataset, its median contribution to local species richness (that is, species of a phylum as a percentage of all species in a grid cell) is 81%, followed by Mollusca (8%), Arthropoda (7%), and Echinodermata (3%) (Extended Data Fig. 5)*”. Therefore, local species richness estimates are dominated by Chordata (and consequently fishes) despite a somewhat balanced representation of vertebrates and invertebrates in terms of the number of species in the database.

To account for this disparity among taxa we conducted new analyses of exposure and opportunity at the level of phyla. The findings are presented in lines 171-177. “*Spatio-temporal patterns of opportunity and exposure are largely consistent across phyla, with opportunities projected to arise mainly in temperate and polar regions and exposure to occur in the tropics (Extended Data Figs. 3 and 4). Chordates account for the highest proportion of the projected exposures and opportunities in most grid cells, followed by molluscs and arthropods (Extended Data Figs. 3-5). In general, taxa that make up a higher proportion of the species richness in an assemblage contribute more to the magnitude of opportunities and exposure (Extended Data Fig. 5).*”

The other issue I see related to the biological aspect is the maximum dispersal distance. Attributing a single value to all taxa seems problematic. I appreciate that the authors tested both 10 and 50 km/year. I think 10 is much more realistic and that 50 is likely a large overestimate, but it's good to see congruency between both values. However, if 50 km/year is near the maximum value observed for the most mobile taxa, I would suspect this leads to a strong over-estimation. For instance, if the database contains many sessile organisms with limited larval dispersal (e.g., sponges), those taxa are certainly not shifting at 50 km/year. Would it be possible to calculate unique dispersal values for different taxa? Why not allow sponges, fishes, seaweeds, etc. to all have their own dispersal abilities, this would lead to a much more realistic projection.

A16: We acknowledge that a dispersal rate of 50 km year⁻¹ likely overestimates the number of thermal opportunities for many species, and that using rates specific to individual taxa would result in more realistic estimates. However, since fishes drive most opportunities, incorporating such dispersal rates for fish in general would have minimal impact on our projections unless we added dispersal estimates for individual genera or species, which would be very challenging due to the limited taxonomic coverage of data in species range shift and dispersal databases. Moreover, according to BioShifts (Lenoir et al. 2020)², the largest species range shift database available to our knowledge, the mean velocity of range shifts is very similar for higher taxonomic levels with marine Actinopterygii, mollusks, and arthropods all approximately 14 km year⁻¹ (Fig. 3a in Lenoir et al. 2020). Therefore, even if we added dispersal rates specific to higher taxonomic groups, the rates for most species would remain similar. The 14 km year⁻¹ estimate from Lenoir et al. 2020 is also similar to the 10 km year⁻¹ used in our analysis.

Nevertheless, we agree that a better approach would be to focus on the more realistic dispersal estimates of 10km year⁻¹ rather than those obtained using the maximum dispersal rates. Therefore, we shifted the emphasis of the main text to

² Lenoir et al. (2020) Species better track climate warming in the oceans than on land. *Nature Ecology & Evolution* 4, 1044-1059.

the results derived from a dispersal rate of 10 km year⁻¹, aligning closely with the median rate for fishes and other groups in the BioShifts database. This adjustment yields a more realistic and conservative estimate of thermal opportunities. We relocated the results based on a 50 km year⁻¹ dispersal rate to the extended data and now refer to them as an upper limit for opportunity emergence.

Those are my major comments. I would like to see the reviewers consider these thoughts and minimally i) provide information on the representation of different taxa in the data, and ii) discuss these implications of their paper. Beyond that I have no major recommendations as I find the paper is exceptional in all aspects.

A17: Thank you for the positive comment and please see the responses above.

Minor and line-by-line comments below:

Abstract – if you have room, I would recommend defining exposure and opportunities in the abstract.

A18: Definitions added as suggested.

Abstract – I think you could be more precise here – there are many studies on temporal biodiversity projections, but most are for a single time period, so annually-resolved or continuous assessments are what is lacking.

A19: We edited the abstract and removed this sentence from it.

Line 66 – hourly temperatures? Is this really defined on an hourly time scale?

A20: We used the definition provided in the original study, but we agree that it could be confusing. The sentence now reads "the smaller difference between body temperatures and the upper critical thermal limit of a species" (lines 53-54).

Line 154: basing this statistic on 5% of species richness seems a bit low, no? 10% or higher would be more powerful? I think it's reasonable to expect a change in 5% of current richness almost everywhere, so it is not a very "big" or surprising result?

A21: We are now basing the results on 10%. Figure 1 now includes both statistics 5% and 10%.

Figure 2 – a dashed or solid black horizontal line at 0 could help emphasize.

A22: We added a white line at 0, to avoid it being confused with the red or dark blue of the polar regions.

Discussion – I think you should speak about other potential limiting factors like competition, habitat suitability, or other biotic and abiotic conditions that might limit species from taking advantage of opportunities that arise.

A23: We added this in the Discussion (lines 416-427).

Reviewed by:

Matthew McLean

[Editorial note: contact details redacted]

Reviewer #3:

This manuscript calculates two things for the global oceans: first, how many of the species currently modeled to occupy any given grid cell will have their thermal niches exceeded under different climate futures (and when); and second, when and where thermally suitable climates will arise for the same species in adjacent locations. There are two things I especially like about this paper. The first is that it quantifies possible habitat losses on the same footing as possible habitat gains, which is a bit tricky to do—colonization processes happen very differently from extinction processes; the authors circumvent this difficulty by focusing just on thermal habitat, not actual species distributions. Relatedly, the second thing I want to compliment is the careful way the authors present their analysis and results in the context of changing thermal habitat suitability rather than projected changing species distributions. This is a chronic issue in global change biology, when researchers conflate a changing climate with changing biogeography without first testing that hypothesis; I really appreciate how this paper emphasizes that thermal exposure and opportunity set some bounds on possible future species shifts but do not in themselves represent biodiversity projections.

A24: We thank the Reviewer for the positive feedback.

I should note that it is impossible to fully review this paper because it does not comply with the Nature Communications policy that peer review of code is required when the code is central to manuscript results, as in this case (see <https://www.nature.com/nature-portfolio/editorial-policies/reporting-standards#availability-of-computer-code>). The manuscript is also missing statements on Data Availability and Code Availability. Thus, the Reporting Summary for this submission should be viewed as incomplete. That said, I have provided as complete a review as I can of the manuscript alone.

A25: We have now included a data availability and code availability section in the manuscript (lines 702-709). The code and post-processed used in our analyses are now available here: <https://doi.org/10.6084/m9.figshare.25197446.v1>.

The methods are relatively straightforward and well-described. They also build on existing studies, including some by the authors themselves. Indeed, given that the entire “exposure” half of the analysis appears to be virtually identical to Pigot et al. 2023, I was surprised to not see that manuscript cited and discussed here. I imagine this is partly due to the journal’s constraint on number of citations; the authors should feel free to add a reference to and discussion of Pigot et al. 2023, and blame this Reviewer if it puts them over the recommended number of references!

A26: We have now included the reference in the text (line 76).

Like many analyses of global marine biogeography, this project used Aquamaps. Aquamaps is both the most convenient tool for this purpose and highly limited in its accuracy and applicability. The authors may be aware of other options for mitigating this, but my understanding based on O’Hara et al. 2017 is that best practices include testing how sensitive the results are of studies using Aquamaps by (1) using IUCN range maps for comparison and/or (2) subsetting Aquamaps to only the species whose range maps are expert-reviewed. Since the authors of the present study had fairly relaxed criteria for which taxa to include (only 10 occurrence points used to model the entire range), I think an analysis of the sensitivity of these results to Aquamaps’ often-erroneous range estimation algorithm would be appropriate. (Another issue: The authors acknowledge this when discussing the different probability of occurrence thresholds, e.g. in Extended Data Fig 5, but a different point I should raise about Aquamaps is that it models marine species’ ranges as being absolutely huge; this is a problem when Aquamaps is used in studies like this one because it models vast swaths of the oceans as being part of any given species’ range when that species has never been found there. The authors ratcheted down the threshold for probability of occurrence in their analysis, from 0.5 to 0. I probably would have ratcheted it up to a much higher value to make it more likely that the modeled ranges are reflective of where species can actually be found.)

A27: In order to investigate whether Aquamaps is overestimating species ranges, we carried out two additional analyses. First, we compared species range sizes obtained from Aquamaps and IUCN. To this end, we downloaded the latest

version of IUCN range maps for marine fishes, the most species-rich group in our analysis. The dataset comprises 4,189 species, of which 2,510 are present in the Aquamaps dataset. For each species, we compared the range sizes by dividing the logarithm of the range size obtained from Aquamaps by the logarithm of the range size from IUCN. If the result is > 1 , the range size from IUCN is larger than the range size from Aquamaps. Across this sample, the difference in range sizes was relatively small, showing that IUCN and Aquamaps provide comparable estimates of range sizes for this group. IUCN range sizes were generally slightly smaller compared to Aquamaps with a 0.0 probability of occurrence threshold (median ratio 0.97), but IUCN ranges are generally slightly larger when compared to Aquamaps with a 0.5 probability threshold (median ratio 1.02). Given we present our results using Aquamaps species' ranges with a 0.5 probability threshold this additional analysis shows that with respect to range size we have taken a relatively conservative approach in our analysis that is in line with and slightly more conservative than IUCN, and thus we did not conduct further analysis with higher Aquamaps thresholds.

Additionally, we compared the spatial overlap between ranges, assessing whether Aquamaps and IUCN estimate the range in the same grid cells. We calculated the overlap by determining the shared grid cells between the two datasets, then dividing the number of shared grid cells by the total number of unique grid cells from both datasets.

The median range overlap between both datasets was $>99\%$ for both Aquamaps with the 0.0 and 0.5 probability threshold. These analyses show that IUCN and Aquamaps provide comparable estimates of species ranges. Therefore, we believe that a further sensitivity analysis using a higher probability of occurrence in Aquamaps that is > 0.5 is not required, since for both datasets species ranges are very similar. We have added text on these additional analyses and comparisons to IUCN in the Methods (lines 546-562) and present the results of the analysis in Supplementary Fig. 1.

One technical question: Am I correct in interpreting L546-548 to mean that from one year to the next, thermal opportunity was calculated for every species within an area representing how far they could be expected to disperse (based on average or maximum dispersal distances) in 85 years? If the focus of this manuscript is the temporal dynamics of changing climates, it seems puzzling to assume each species' has access to 85 years' worth of dispersal distance in each year. I understand long-distance dispersal is a factor but this is not a particularly direct way to quantify its potential impacts; indeed these distances are so large even the smaller value (850 km), if considered an upper limit for a single year, essentially represents no barriers to dispersal. It might be more realistic to re-run the annual projections giving each species the authors' own estimates of dispersal distance per year (10-50 km) to see how it would constrain the projections.

A28: The reviewer is interpreting our analysis correctly. We allowed opportunities to emerge anywhere within the buffer estimated based on either a 10 km year⁻¹ or 50 km year⁻¹ dispersion. This means that for a 10 km year⁻¹ rate, opportunities can emerge anywhere on a radius of approximately 850 km around the current range of the species. As this Reviewer highlighted in their earlier comments we emphasise in the manuscript that our study does not attempt to model the process of range expansions and species distributions, but focuses instead on thermal habitat dynamics. Range expansion dynamics are highly stochastic, including complex ecological and evolutionary processes and may be dominated by rare long distance dispersal events. Here we aim to address a simpler question (while still considering potential for cumulative dispersal and rare long distance dispersal events): how will thermal opportunities arise within the region where a species could potentially disperse to this century?

We selected our approach because our study is focused on projecting the future dynamics of the emergence of thermal opportunities (including their persistence or transience over this century) that may influence species range dynamics. We therefore took a more permissive approach to identifying such thermal opportunities within a dispersal constrained buffer zone rather than a more restrictive approach with annual dispersal increments because this allows us to

identify the temporal dynamics of opportunities that may arise far from the current range edge, but could be reached by rare long distance dispersal events or if the opportunities persist for several decades. Furthermore, our study shows that between 76%–97% opportunities are projected to persist until the end of the century. Therefore, even if an opportunity arises further from the current range of the species, that opportunity is likely to remain open until 2100 and since the species can reach any site within the buffer by 2100, the species would have enough time to reach all opportunities that persist until that time. Therefore, using an incremental dispersal approach, while also computationally intensive, would have limited impacts on the patterns of thermal opportunity emergence identified in our study. We have included in the manuscript a brief discussion of these issues (lines 401-415).

Regarding assigning species-specific dispersal rates: this point was also raised by Reviewer 2. We would like to refer the Reviewer to response *A16*, where we discuss why adding high-level taxa-specific dispersal rates would have little impact on our findings and is limited by data availability for lower taxonomic levels.

I commend the authors for analyzing multiple climate scenarios, which numerous recent articles have argued is key for maximizing applicability of climate and biodiversity research to e.g. the IPCC process (Schoeman et al. 2023). However, the authors offer no explanation for the choice to present main text results based on SSP5-8.5, the high-emissions, no-mitigation scenario that is generally considered to be an unlikely future (Hausfather & Peters 2020). Actually, unless I missed it, the choice of scenarios and what each mean for the future is not really introduced anywhere in this manuscript. I'd encourage the authors to add a few sentences to the Introduction along these lines, because in any manuscript talking about the future, it's important to introduce which potential future we're talking about and why. The manuscript text implies that the main results analyzed a likely future because it uses the word "will" frequently in reference to the modeled projections. Given that, I suggest that the authors re-center their results around SSP2-4.5 (currently considered the most probable of their three scenarios), or re-frame the paper to explain why they aimed to illustrate a worst-case scenario for the

oceans that is thankfully unlikely to come to pass. Presenting all three scenarios in-text, which the authors do for some results (e.g., Fig. 2), is another option. Guidance for choosing and discussing climate scenarios that may be helpful can be found in Burgess et al. 2023.

A29: We agree with the Reviewer that presenting the findings from all three scenarios would improve the manuscript. All our figures now include the results of all three SSPs. We have also reworded the text to reduce emphasis on SSP5-8.5.

L562: This sentence doesn't make sense to me (it seems to say "when temperature becomes unsuitable, we classify that cell as suitable in the future"). Is "outside" meant to be "within"? (It also doesn't seem to follow from the previous sentence, which I interpret to say "when our model said temperature was suitable but Aquamaps said the cell was uninhabitable, we believed Aquamaps because it uses other non-temperature parameters in its model.")

A30: We were referring to cells within the current range of the species, not cells that were excluded due to niche unfilling. We thank the Reviewer for picking this up. We have revised the sentence to read "Opportunities could also arise for grid cells within the current range of the species, but only if an exposure event has occurred previously in the grid cells" on lines 623-625.

REVIEWERS' COMMENTS

Reviewer #1 (Remarks to the Author):

The authors have done an excellent job by incorporating the suggestions and clarifications that were made. I have no further comments on the current version. Congratulations

Reviewer #2 (Remarks to the Author):

The authors have successfully addressed all my concerns. They did a very thorough job. My recommendation is to accept and publish.